



# Ground-based validation of the Copernicus Sentinel-5p TROPOMI $NO_2$ measurements with the NDACC ZSL-DOAS, MAX-DOAS and Pandonia global networks

Tijl Verhoelst[1], Steven Compernolle[1], Gaia Pinardi[1], Jean-Christopher Lambert[1], Henk J. Eskes[2], Kai-Uwe Eichmann[3], Ann Mari Fjæraa[4], José Granville[1], Sander Niemeijer[5], Alexander Cede[6,7], Martin Tiefengraber[7], François Hendrick[1], Andrea Pazmiño[8], Alkiviadis Bais[9], Ariane Bazureau[8], K. Folkert Boersma[2,10], Kristof Bognar[11], Angelika Dehn[12], Sebastian Donner[13], Aleksandr Elokhov[14], Manuel Gebetsberger[7], Florence Goutail[8], Michel Grutter de la Mora[15], Aleksandr Gruzdev[14], Myrto Gratsea[16], Georg H. Hansen[17], Hitoshi Irie[18], Nis Jepsen[19], Yugo Kanaya[20], Dimitris Karagkiozidis[9], Rigel Kivi[21], Karin Kreher[22], Pieternel F. Levelt[2,23], Cheng Liu[24], Moritz Müller[7], Monica Navarro Comas[25], Ankie J.M. Piters[2], Jean-Pierre Pommereau[8], Thierry Portafaix[26], Olga Puentedura[25], Richard Querel[27], Julia Remmers[13], Andreas Richter[3], John Rimmer[28], Claudia Rivera Cárdenas[15], Lidia Saavedra de Miguel[12], Valery P. Sinyakov[29], Kimberley Strong[11], Michel Van Roozendael[1], J. Pepijn Veefkind[2], Thomas Wagner[11], Folkard Wittrock[3], Margarita Yela González[22], and Claus Zehner[10]

[1]Royal Belgian Institute for Space Aeronomy (BIRA-IASB), Ringlaan 3, 1180 Uccle, Belgium
[2]Royal Netherlands Meteorological Institute (KNMI), Utrechtseweg 297, 3730 AE De Bilt, The Netherlands
[3]Institute of Environmental Physics (IUP), University of Bremen, Otto-Hahn-Allee 1, D-28359 Bremen, Germany
[4]Norsk Institutt for Luftforskning (NILU), Instituttveien 18, 2007 Kjeller, Norway
[5]Science [&] Technology Corporation (S[&]T), Delft, The Netherlands
[6]Goddard Space Flight Center (NASA/GSFC), Greenbelt, MD, USA
[7]LuftBlick, Kreith, Austria & Institute of Meteorology and Geophysics, University of Innsbruck, Innsbruck, Austria
[8]Laboratoire Atmosphères, Milieux, Observations Spatiales (LATMOS), UVSQ/UPMC/CNRS, Guyancourt, France
[9]Laboratory of Atmospheric Physics, Aristotle University of Thessaloniki (AUTH), Thessaloniki, Greece
[10]Meteorology and Air Quality group, Wageningen University, 6700 AA Wageningen, The Netherlands
[11]Department of Physics, University of Toronto, 60 St. George Street, Toronto, Ontario, M5S 1A7, Canada
[12]European Space Agency/Centre for Earth Observation (ESA/ESRIN), Frascati, Italy
[13]Max-Planck-Institut für Chemie (MPI-C), Hahn-Meitner-Weg 1, 55128 Mainz, Germany
[14]A.M. Obukhov Institute of Atmospheric Physics (IAP), Russian Academy of Sciences, Moscow, Russian Federation
[15]Centro de Ciencias de la Atmósfera, Universidad Nacional Autónoma de México (UNAM), Mexico City, Mexico
[16]National Observatory of Athens, Lofos Nymphon - Thissio, PO Box 20048 - 11810, Athens, Greece
[17]Norsk Institutt for Luftforskning (NILU), P.O. Box 6606 Langnes, NO-9296 Tromsø, Norway
[18]Center for Environmental Remote Sensing, Chiba University (ChibaU), Chiba, Japan
[19]Danish Meteorological Institute (DMI), Lyngbyvej 100, 2100 Copenhagen, Denmark
[20]Research Institute for Global Change (JAMSTEC), Yokohama, Japan
[21]Space and Earth Observation Centre, Finnish Meteorological Institute, Tähteläntie 62, FI-99600 Sodankylä, Finland
[22]BK Scientific GmbH, Astheimerweg 42, 55130 Mainz, Germany
[23]University of Technology Delft, Mekelweg 5, 2628 CD Delft, The Netherlands
[24]Department of Precision Machinery and Precision Instrumentation, University of Science and Technology of China, Hefei, 230026, China
[25]Atmospheric Research and Instrumentation, National Institute for Aerospace Technology (INTA), Madrid, 28850, Spain





[26]Laboratoire de l'Atmosphère et des Cyclones (LACy), Université de La Réunion, Saint-Denis, France
[27]National Institute of Water and Atmospheric Research (NIWA), Private Bag 50061, Omakau, Central Otago, New Zealand
[28]University of Manchester, Oxford Rd, M13 9PL Manchester, United Kingdom
[29]Kyrgyz National University of Jusup Balasagyn (KNU), 547 Frunze Str., Bishkek, Kyrgyz Republic

**Correspondence:** Tijl Verhoelst (tijl.verhoelst@aeronomie.be)

**Abstract.** This paper reports on consolidated ground-based validation results of the atmospheric $NO_2$ data produced operationally since April 2018 by the TROPOMI instrument on board of the ESA/EU Copernicus Sentinel-5 Precursor (S5p) satellite. Tropospheric, stratospheric, and total $NO_2$ column data from S5p are compared to correlative measurements collected from, respectively, 19 Multi-Axis DOAS (MAX-DOAS), 26 NDACC Zenith-Scattered-Light DOAS (ZSL-DOAS), and 25

PGN/Pandora instruments distributed globally. The validation methodology gives special care to minimizing mismatch errors due to imperfect spatio-temporal co-location of the satellite and correlative data, e.g., by using tailored observation operators to account for differences in smoothing and in sampling of atmospheric structures and variability, and photochemical modelling to reduce diurnal cycle effects. Compared to the ground-based measurements, S5p data show, on an average: (i) a negative bias for the tropospheric column data, of typically -23 to -37% in clean to slightly polluted conditions, but reaching values as high

as $-51\%$ over highly polluted areas; (ii) a slight negative bias for the stratospheric column data, of about -0.2 Pmolec/cm$^2$, i.e. approx. -2% in summer to -15% in winter; and (iii) a bias ranging from zero to $-50\%$ for the total column data, found to depend on the amplitude of the total $NO_2$ column, with small to slightly positive bias values for columns below 6 Pmolec/cm$^2$ and negative values above. The dispersion between S5p and correlative measurements contains mostly random components, which remain within mission requirements for the stratospheric column data (0.5 Pmolec/cm$^2$), but exceed those for the tropo-

spheric column data (0.7 Pmolec/cm$^2$). While a part of the biases and dispersion may be due to representativeness differences, it is known that errors in the S5p tropospheric columns exist due to shortcomings in the (horizontally coarse) a-priori profile representation in the TM5-MP chemistry transport model used in the S5p retrieval, and to a lesser extent, to the treatment of cloud effects. Although considerable differences (up to 2 Pmolec/cm$^2$ and more) are observed at single ground-pixel level, the near-real-time (NRTI) and off-line (OFFL) versions of the S5p $NO_2$ operational data processor provide similar $NO_2$ column

values and validation results when globally averaged, with the NRTI values being on average 0.79% larger than the OFFL values.

# 1 Introduction

Nitrogen oxides, and in particular the NOx (NO and $NO_2$), are important trace gases both in the troposphere and the strato-

sphere. In the troposphere they are produced mainly by the combustion of fossil and other organic fuels, and by the production

and use of nitrogen fertilizers for agriculture. They can also have a natural origin, e.g., lightning, biological processes in soils, and biomass burning. The NO/$NO_2$ ratio varies with solar illumination primarily, from 0.2-0.5 during the day down to zero at night. NOx are converted to nitric acid and nitrates which are removed by dry deposition and rain, resulting in a tropospheric lifetime of a few hours to days. Tropospheric NOx are pollutants as well as proxies for other pollutants resulting from the





(high-temperature) combustion of organic fuels. They are precursors for tropospheric ozone and aerosols and contribute to acid
rain and smog. Because of their adverse health effects, local to national regulations to limit boundary layer NOx concentrations are now in place in a long list of countries across the world. In the stratosphere NOx are formed by the photolosysis of tropospheric nitrous oxide ($N_2O$) produced by biogenic and anthropogenic processes and going up through the troposphere and stratosphere. Stratospheric NOx controls the abundance of ozone, as a catalyst in ozone destruction processes, but also by mitigating ozone losses caused by catalytic cycles involving anthropogenic halogens through the lock-up of these halogens in
so-called long-lived reservoirs.

The global distribution, cycles and trends of atmospheric $NO_2$ have been measured from space by a large number of instruments on low-Earth orbit (LEO) satellites. Since the late 1970s, its stratospheric and sometimes mesospheric abundance has been measured by limb viewing and solar occultation instruments working in the UV-visible and infrared spectral ranges: SME, LIMS, SAGE(-II), HALOE, POAM-2/3... and, in the last decade, OSIRIS, GOMOS, MIPAS, SCIAMACHY, Scisat ACE, and
SAGE-III. Follow-on missions combining limb and occultation measurements are in development, like ALTIUS planned for the coming years. Pioneered in 1995 with ERS-2 GOME (Burrows et al., 1999), which for the first time brought into space $NO_2$ column measurements by Differential Optical Absorption Spectroscopy (DOAS, Noxon et al. (1979); Platt and Perner (1983)), the global monitoring of tropospheric $NO_2$ has continued uninterruptedly with a suite of UV-visible DOAS instruments with improving sensitivity and horizontal resolution: Envisat SCIAMACHY (Bovensmann et al., 1999), EOS-Aura OMI (Levelt
et al., 2018), and the series of MetOp-A/B/C GOME-2 (Valks et al., 2011; Liu et al., 2019b).

Owing to its cardinal role in air quality, tropospheric chemistry, stratospheric ozone, and as a precursor of essential climate variables (ECV), the monitoring of atmospheric $NO_2$ on the global scale has been given proper attention in the European Earth Observation programme Copernicus. The Copernicus Space Component (CSC) develops a constellation of atmospheric composition Sentinel satellites with complementary $NO_2$ measurement capabilities, consisting of Sentinel-4 geostationary mis-
sions (with hourly monitoring over Europe) and Sentinel-5 LEO missions (with daily monitoring globally) to be launched from 2023 on-wards. A $NO_2$ measurement channel is also planned for the Copernicus Carbon Dioxide Monitoring mission CO2M for better attribution of the atmospheric emissions. First element in orbit of this LEO+GEO constellation, the TROPOspheric Monitoring Instrument (TROPOMI) was launched on board of ESA's Sentinel-5 Precursor (S5p) early-afternoon LEO satellite in October 2017. This hyperspectral imaging spectrometer measures the Earth radiance, at 0.2-0.4 nm resolution in the visible
absorption band of $NO_2$, over ground pixels as small as $7.0 \times 3.5 km^2$ or $5.5 \times 3.5 km^2$ (before and after the switch to smaller pixel size on August 6, 2019, respectively) with an almost daily global coverage thanks to a swath width of $2600 \, km$.

Pre-launch mission requirements for the Copernicus Sentinel $NO_2$ data are, for the tropospheric $NO_2$ column, a bias lower than 50% and an uncertainty lower than 0.7 Pmolec/cm$^2$, and for the stratospheric $NO_2$ column, a bias lower than 10% and an uncertainty lower than 0.5 Pmolec/cm$^2$ (ESA, 2017a, b). Since the beginning of its nominal operation in April 2018, in-
flight compliance of S5p TROPOMI with these mission requirements has been monitored routinely by means of comparisons to ground-based reference measurements in the Validation Data Analysis Facility (VDAF) of the S5p Mission Performance Centre (MPC) and by confrontation with similar satellite data from OMI and GOME-2. The Copernicus S5p MPC routine operations validation service is complemented with ground-based validation studies carried out in the framework of ESA's



S5P Validation Team (S5PVT) through research projects funded nationally like NIDFORVAL (see details in the Acknowl-
edgments section). Ground-based validation of satellite $NO_2$ data (e.g., Petritoli et al., 2003; Brinksma et al., 2008; Celarier
et al., 2008; Ionov et al., 2008; Valks et al., 2011; Compernolle et al., 2020b; Pinardi et al., 2020) relies classically on three
types of UV-visible DOAS instruments which, thanks to complementary measurement techniques, provide all together cor-
relative observations sensitive to the three components of the S5p data product: Multi-Axis Differential Optical Absorption
Spectroscopy (MAX-DOAS) measures the tropospheric column during the day, Zenith-Scattered-Light DOAS (ZSL-DOAS)
the stratospheric column at dawn and dusk, and Pandora direct Sun instruments the total column during the day, respectively.
Currently, those three types of instruments contribute to global monitoring networks. Fig. 1 shows the geographical distribution
of instruments contributing data to the reported S5p validation study.

In this paper, we report on the consolidated results of the S5p $NO_2$ ground-based validation activities for the first two
years of nominal operation. The TROPOMI tropospheric, stratospheric and total column data products under investigation are
described in Sect. 2, with a brief assessment of the coherence between the data generated by the near-real-time (NRTI) and
off-line (OFFL) channels of the operational processors. For clarity, we present in separate sections results for the stratospheric
(Sect. 3), tropospheric (Sect. 4) and total (Sect. 5) $NO_2$ columns. These three sections include the description of the S5p data
preparation, of the ground-based validation data, of the preparation of the filtered, co-located, and harmonized data pairs to be
compared, and the comparison results. Robust, harmonised statistical estimators are derived from the comparisons consistently
throughout the paper: the median difference as a proxy for the bias, and half of the 68% interpercentile (IP68/2) as a measure
of the comparison spread (equivalent to a standard deviation for a Normal distribution, but much less sensitive to unavoidable
outliers). Thereafter, in Sect. 6, these individual results are assembled and discussed all together, to derive conclusions on their
mutual coherence, on the fitness-for-purpose of the S5p data, and on remaining challenges for the accurate validation of $NO_2$
observations from space.

## 2 S5p TROPOMI data

### 2.1 Data description and filtering

The retrieval of $NO_2$ (sub)columns from TROPOMI Earth nadir radiance and solar irradiance spectra is a 3-step process relying
on DOAS and on a Chemical Transport Model (CTM) based stratosphere-troposphere separation. The TROPOMI $NO_2$ algo-
rithm is an adaptation of the QA4ECV community retrieval approach (Boersma et al. (2018)) and of the DOMINO/TEMIS
algorithm (Boersma et al., 2007, 2011), already applied successfully to heritage and current satellite data records (GOME,
SCIAMACHY, OMI, GOME-2). In the first step, the integrated amount of $NO_2$ along the optical path, or slant column den-
sity (SCD), is derived using the classical DOAS approach (Platt and Perner, 1983). In the second step, the retrieved SCD is
assimilated by the TM5-MP CTM to allocate a vertical profile of the $NO_2$ concentration, needed for the separation between
stratospheric and tropospheric SCDs. This assimilation procedure favours observations over pristine, remote areas where the
entire $NO_2$ SCD can be attributed to the stratospheric component. Assuming relatively slow changes in the stratospheric NOx
field, the model transports information to areas with a more significant tropospheric component. In the third step, the three

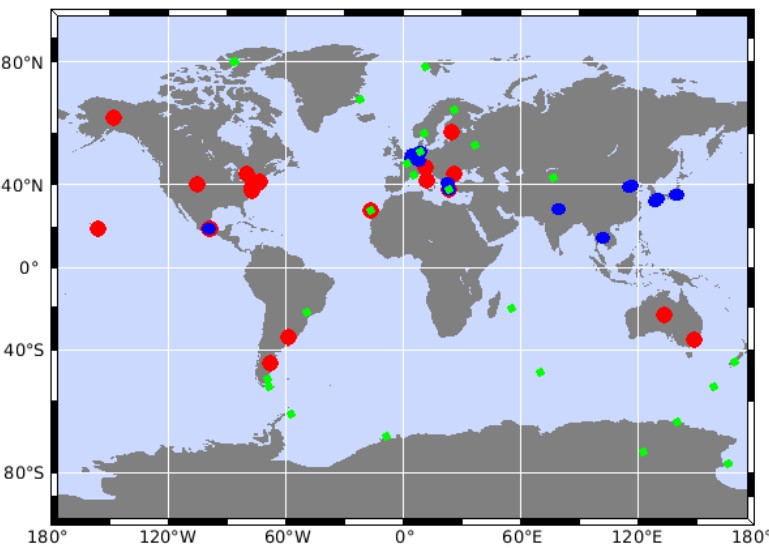

**Figure 1.** Geographical distribution of the UV-visible DOAS spectrometers contributing the ground-based correlative measurements: 26 NDACC ZSL-DOAS instruments in green, 19 MAX-DOAS instruments in blue, and 25 PGN instruments in red.

slant (sub)column densities are converted into vertical (sub)column densities using appropriate Air Mass Factors (AMFs). The CTM can be run either in forecast mode, using 1-day forecast meteorological data from the European Centre for Medium-range Weather Forecasts (ECMWF), or in a more delayed processing mode, using 0-12hour forecast meteorological data. The former

is used for near-real-time (NRTI) processing of the TROPOMI measurements, the latter for the offline (OFFL) production. For full technical details, the reader is referred to the Product Readme File (PRF), Product User Manual (PUM) and Algorithm Theoretical Basis Document (ATBD), all available at http://www.tropomi.eu/data-products/nitrogen-dioxide. A detailed description and quality assessment of the derived slant column data is already published by van Geffen et al. (2020), and a publication on satellite inter-comparison of vertical column data is under preparation (Eskes et al., 2020). The current paper

addresses the independent ground-based validation of vertical sub-column densities in the troposphere and stratosphere and of the vertical total column. The S5p data set validated here covers the nominal operational phase (Phase E2) of the S5p mission, starting in April 2018 and up to February 2020. No data obtained during the commissioning phase of the satellite have been used. Table 1 provides an overview of the processor versions this corresponds to. They constitute as continuous a data set as possible from May (NRTI) or October (OFFL) 2018 onward. Combining RPRO (May-October 2018) with OFFL, a coherent

dataset with version OFFL processor v01.02.02 or higher can be obtained.





**Table 1.** Identification of the S5p $NO_2$ data versions validated here: near-real-time channel (NRTI), off-line channel (OFFL) and interim reprocessing (RPRO). Major updates were those leading to v01.02.00 and to v01.03.00.

| Processor version | start orbit | start date | end orbit | end date |
|---|---|---|---|---|
| NRTI | | | | |
| 01.00.01 | 2955 | 2018-05-09 | 3364 | 2018-06-07 |
| 01.00.02 | 3745 | 2018-07-04 | 3946 | 2018-07-18 |
| 01.01.00 | 3947 | 2018-07-18 | 5333 | 2018-07-24 |
| 01.02.00 | 5336 | 2018-10-24 | 5929 | 2018-12-05 |
| 01.02.02 | 5931 | 2018-12-05 | 7517 | 2019-03-27 |
| 01.03.00 | 7519 | 2019-03-27 | 7999 | 2019-03-30 |
| 01.03.01 | 7999 | 2019-03-30 | 9158 | 2019-07-20 |
| 01.03.02 | 9159 | 2019-07-20 | current version | |
| OFFL | | | | |
| 01.02.00 | 5236 | 2018-10-17 | 5832 | 2018-11-28 |
| 01.02.02 | 5840 | 2018-11-29 | 7424 | 2019-03-20 |
| 01.03.00 | 7425 | 2019-03-20 | 7906 | 2019-04-23 |
| 01.03.01 | 7907 | 2019-04-23 | 8814 | 2019-06-26 |
| 01.03.02 | 8815 | 2019-06-26 | current version | |
| RPRO | | | | |
| 01.02.02 | 2836 | 2018-05-01 | 5235 | 2018-10-17 |

Besides very detailed quality flags, the S5p $NO_2$ data product includes a combined quality assurance value (qa_value) enabling end users to easily filter data for their own purpose. For tropospheric applications (when not using the averaging kernels), the guideline is to use only $NO_2$ data with a qa_value $> 0.75$. This removes very cloudy scenes (cloud radiance fraction $> 0.5$), snow- or ice-covered scenes, and problematic retrievals. For stratospheric applications, where clouds are less of an issue, a more relaxed threshold of qa_value $> 0.5$ is recommended. These data filtering recommendations have been applied here, where the stricter requirement of qa_value $> 0.75$ has been used for the total column validation as well. Again, further details on this can be found in the PRF, PUM, and ATBD.

## 2.2 Mutual coherence between NRTI and OFFL

As described in Sect. 2.1, the main difference between the NRTI and OFFL data processors lies in the use of either forecast or analysis ECMWF meteorological data as input, and consequently the use of either forecast or analysis TM5-MP vertical $NO_2$ profiles. The mutual consistency between the NRTI and OFFL data products is monitored routinely using data and tools provided by the S5p MPC Level-2 Quality Control Portal (https://mpc-l2.tropomi.eu). Fig. 2 shows that, looking at global means of the $NO_2$ total column, the NRTI and OFFL data look very much alike, with NRTI column values on an average





0.79% larger than those obtained in OFFL. Eight NRTI and six OFFL processor versions are used in this comparison (as
identified in Table 1). The activation of the successive processor versions and the switch to the smaller ground pixel size (on
August 6, 2019) are marked by the yellow vertical lines. As expected both NRTI and OFFL channels show $NO_2$ maxima in
the winter/summer seasons (December, June) and minima near the equinoxes. The scatter also exhibits a seasonal cycle, with
largest values observed in the Northern Hemisphere winter season.

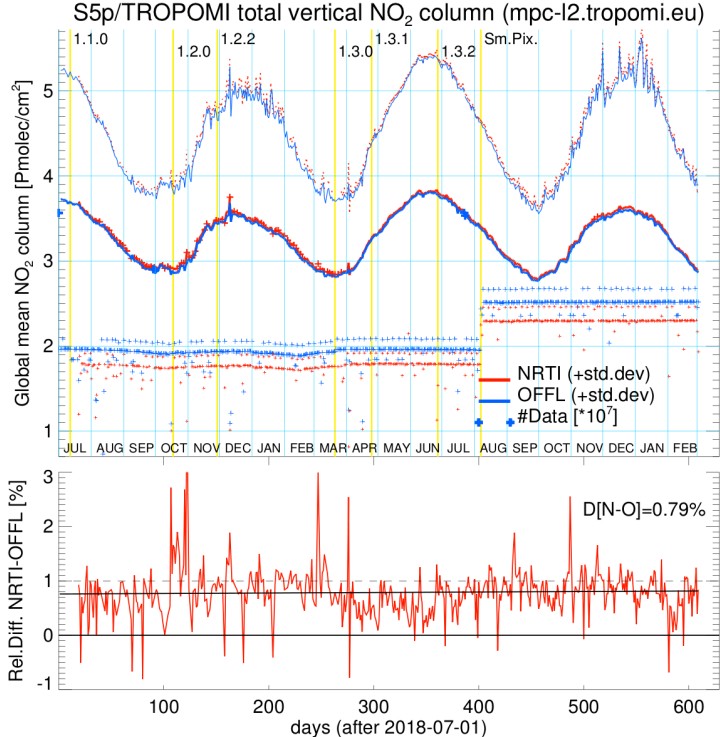

**Figure 2.** *Upper panel:* Time series of the global means of $NO_2$ total column data retrieved with the NRTI (red line) and OFFL (blue line)
processors, and their standard deviation, in Pmolec/cm$^2$, from July 2018 till February 2020. Crosses depict the number of measurements
divided by $10^7$, with the same colour code: red for NRTI, blue for OFFL. Yellow vertical lines indicate the transition dates for processor
upgrades and the switch to the smaller ground pixel size. *Lower panel:* Percent relative difference between NRTI and OFFL global means of
total $NO_2$ values. The Theil-Sen linear regression line (black) is superimposed.

To further assess similarities and differences between the NRTI and OFFL processing channels, $NO_2$ values along individual
orbits are also compared directly. An illustration is given in Figure 3 for S5p orbit # 07407, a randomly selected orbit crossing
Western Europe on a relatively cloud-free day (March 19, 2019).

The three maps of Figure 3 show the difference between NRTI and OFFL values for the total, stratospheric and tropospheric
$NO_2$ column, respectively, together with the corresponding Pearson correlation coefficient and root-mean-square deviation
(RMSD). While the correlation coefficient is high (typically around 0.97), the maps do reveal regions where significant de-



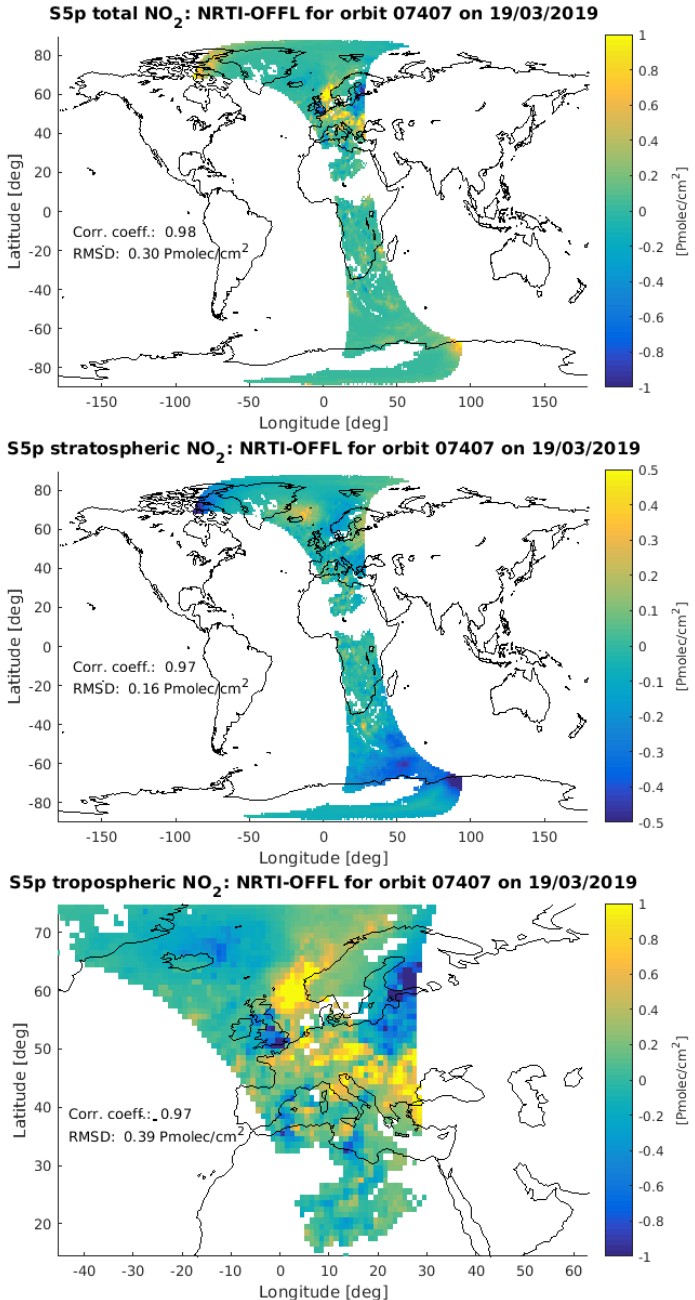

**Figure 3.** Maps of the difference between the NRTI and OFFL $NO_2$ data values for S5p orbit #07407 on March 19, 2019. *Upper panel:* difference between total column values; *Middle panel:* stratospheric columns; *Lower panel:* zoom on the difference in tropospheric column values over Western Europe.





viations occur, up to $\pm 0.5\,\mathrm{Pmolec/cm^2}$ between the NRTI and OFFL stratospheric columns, and up to $\pm 2\,\mathrm{Pmolec/cm^2}$ for both the tropospheric columns and the total columns. Significant differences over South-East England (London) and in the Manchester-Liverpool area are particularly striking. North-East of Iceland, NRTI-OFFL differences in stratospheric and in tropospheric columns are of opposite sign, indicating a different stratosphere/troposphere separation after the slant column retrieval leading to little difference in the total columns. A more detailed investigation targeted solely at regions and times of

significant deviations between NRTI and OFFL would be needed to better reveal the full benefit of the OFFL analysis, but that is beyond the scope of the current paper. What needs to be underlined is that the ground-based validation studies on which the present consolidated results are based upon, do not yield significantly different conclusions for the two processing modes. Therefore, all results reported in this paper may be considered as applicable to the two processing channels.

## 3  Stratospheric column validation

### 3.1  NDACC Zenith-sky DOAS data

Since the pioneering ages of $NO_2$ column measurements from space with ERS-2 GOME in the mid-1990s, ground-based UV-visible DOAS measurements at twilight have served as a reference for the validation of $NO_2$ total column data over un-polluted stations and of $NO_2$ stratospheric column data from all nadir UV-visible satellites to date (e.g., Lambert et al., 1997a, b; Petritoli et al., 2003; Celarier et al., 2008; Ionov et al., 2008; Gruzdev and Elokhov, 2010; Dirksen et al., 2011; Hendrick

et al., 2011; Robles-Gonzalez et al., 2016). Here as well, S5p TROPOMI stratospheric $NO_2$ column data are compared to the correlative measurements acquired by ZSL-DOAS (Zenith-Scattered Light Differential Optical Absorption Spectroscopy) UV-Visible spectrometers (e.g. Solomon et al., 1987; Hendrick et al., 2011, and references therein). A key property of zenith-sky measurements at twilight is the geometrical enhancement of the optical path in the stratosphere (Solomon et al., 1987), which offers high sensitivity to stratospheric absorbers of visible radiation and lower sensitivity to clouds and tropospheric

species (except in the case of strong pollution events during thunderstorms or thick haze, see e.g. Pfeilsticker et al. (1999)). However, the geometrical enhancement also implies horizontal smoothing of the measured information over hundreds of kilo-metres, which requires appropriate co-location methods to avoid large discrepancies with the higher-resolution measurements of TROPOMI, as discussed in Sect. 3.2. Various ZSL-DOAS UV-visible instruments with standard operating procedures and harmonized retrieval methods perform network operation in the framework of the Network for the Detection of Atmospheric

Composition Change (NDACC, De Mazière et al. (2018)). Part of this, over 15 instruments of the SAOZ design (Système d'Analyse par Observation Zénitale) are distributed worldwide and provide data in near-real-time through the CNRS LAT-MOS_RT Facility (Pommereau and Goutail, 1988). For the current work, ZSL-DOAS validation data have been obtained: (1) through the LATMOS_RT Facility (in nearl-real-time processing mode), (2) from the NDACC Data Host Facility (DHF), and (3) via private communication with the instrument operator. The geographical distribution of these instruments is shown in

Fig. 1 and further details are provided in A1 in the supplement. Measurements are made during twilight, at sunrise and sunset, but only sunset measurements are used here for signal-to-noise reasons (larger $NO_2$ column) and as these happen closer in time to the early-afternoon overpass of S5p. NDACC intercomparison campaigns (Roscoe et al., 1999; Vandaele et al., 2005) con-





clude to an uncertainty of about 4-7% on the slant column density. After conversion of the slant column into a vertical column using a zenith-sky AMF, and for the latest version of the data processing, the uncertainty on the vertical column is estimated

to be the order of 10-14% (Yela et al., 2017; Bognar et al., 2019). A limiting factor comes from the temperature dependence of the $NO_2$ absorption cross-sections used in the DOAS retrieval of the slant column density. Most of the NDACC instruments use cross-sections at a single temperature (220 K), which introduces a seasonal error of up to a few percent at middle and high latitudes.

## 3.2 Co-location and harmonization

To account for effects of the photochemical diurnal cycle of stratospheric $NO_2$, the ZSL-DOAS measurements at sunset are adjusted to the early-afternoon S5p overpass time using a model-based correction factor. The latter is calculated with the PSCBOX 1D stacked-box photochemical model (Errera and Fonteyn, 2001; Hendrick et al., 2004) initiated by daily fields from the SLIMCAT chemistry-transport model (CTM). The amplitude of the adjustment factor is sensitive to the effective SZA assigned to the ZSL-DOAS measurements. It is assumed here to be 89.5° or, during polar day and close to polar night,

the largest or smallest SZA reached, respectively. The uncertainty related to this adjustment is estimated to be of the order of 10%, the main source of uncertainty probably being the effective SZA to assign to the full twilight measurement period. To reduce mismatch errors due to the significant difference in horizontal sensitivity between S5p and ZSL-DOAS measurements, individual TROPOMI $NO_2$ stratospheric column data (in ground pixels at high horizontal sampling) are averaged over the much larger footprint of the air mass to which the ground-based zenith-sky measurement is sensitive, see Lambert et al. (1997b,

ISBN 978-1-4614-3908-0, © Springer New York, 2012) and Verhoelst et al. (2015) for details. Note that, as the TROPOMI stratospheric column is a TM5 output, it's true resolution is actually much lower than the pixel size.

## 3.3 Comparison results

Fig. 4 illustrates the comparison between TROPOMI and ground-based ZSL-DOAS SAOZ $NO_2$ data at the NDACC station at Observatoire de Haute Provence (O.H.P.) in Southern France. The time series reveal a small negative bias for TROPOMI,

which is found to be a common feature across the network, but little seasonal structure. The correlation coefficient is excellent and the histogram of the differences has an almost Gaussian shape.

Comparison results for the entire ZSL-DOAS network are presented in Fig. 5. This figure reveals occasionally larger differences in more difficult co-location conditions (e.g. enhanced variability at the border of the polar vortex) but no impact of the TROPOMI pixel size change on August 6th, 2019. The latter result must be interpreted with care as, for these comparisons,

multiple TROPOMI pixels are averaged over the ZSL-DOAS observation operator before comparison (see Sect. 3.2), and as such any change in the noise statistics of individual pixels will be hidden.

Statistical estimators of the bias and scatter per station are presented in box-whisker plots in Fig. 6, and in tabular form in A1. Across the network, S5p NRTI and OFFL stratospheric $NO_2$ column data are generally lower than the ground-based values by approximately 0.2 Pmolec/cm$^2$, with a station-to-station scatter of this bias of similar magnitude (0.3 Pmolec/cm$^2$).

These numbers are within the mission requirement of a maximum bias of 10% (equivalent to 0.2-0.4 Pmolec/cm$^2$, depending



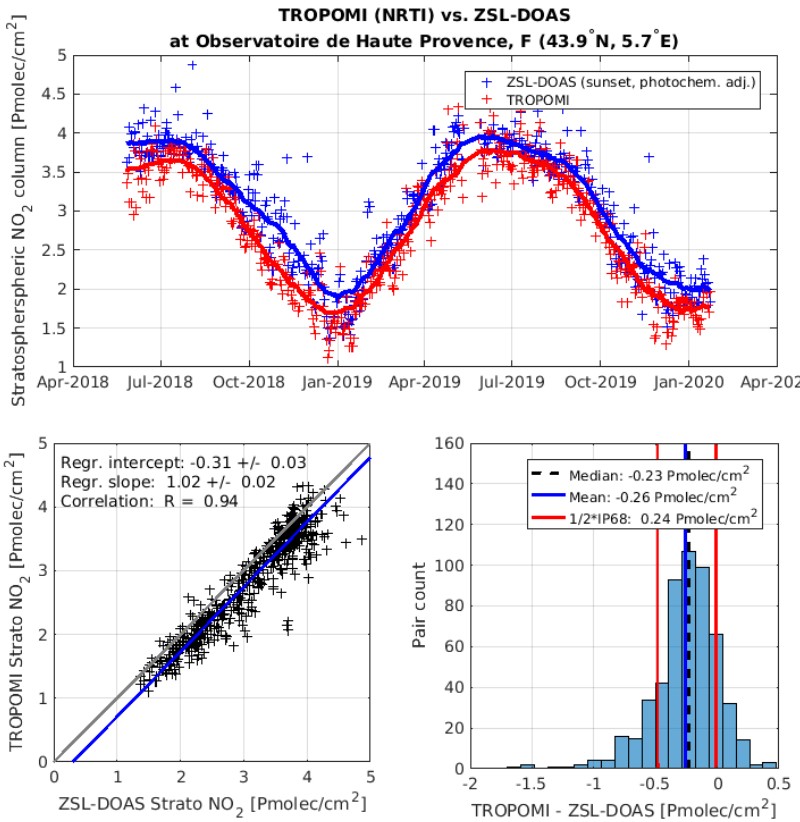

**Figure 4.** *Upper panel:* Time series of S5p NRTI stratospheric $NO_2$ column data co-located with ground-based SAOZ sunset measurements performed by CNRS/LATMOS at the NDACC mid-latitude station of Observatoire de Haute-Provence (France). The latter were adjusted for the photochemical difference between the S5p and twilight solar local times, while S5p data were averaged over the ground-based twilight air mass. Solid lines represent 2-month running medians. *Lower panels:* Scatter plot (left-hand side) and histogram of the difference (right-hand side) with several statistical measures of the agreement between data.

on latitude and season), and within the combined systemic uncertainty of the reference data and their model-based photochemical adjustment. The IP68/2 dispersion of the difference between TROPOMI stratospheric column and correlative data around their median value rarely exceeds 0.3 Pmolec/cm² at sites without tropospheric pollution. When combining random errors in the satellite and reference measurements with irreducible co-location mismatch effects, it can be concluded that the random uncertainty on the S5p stratospheric column measurements falls within mission requirements of max. 0.5 Pmolec/cm² uncertainty.

The potential dependence of the TROPOMI stratospheric column bias and uncertainty on several influence quantities has been evaluated. Fig. 7 shows results for the solar zenith angle (SZA), the fractional cloud cover (CF), and the surface albedo of

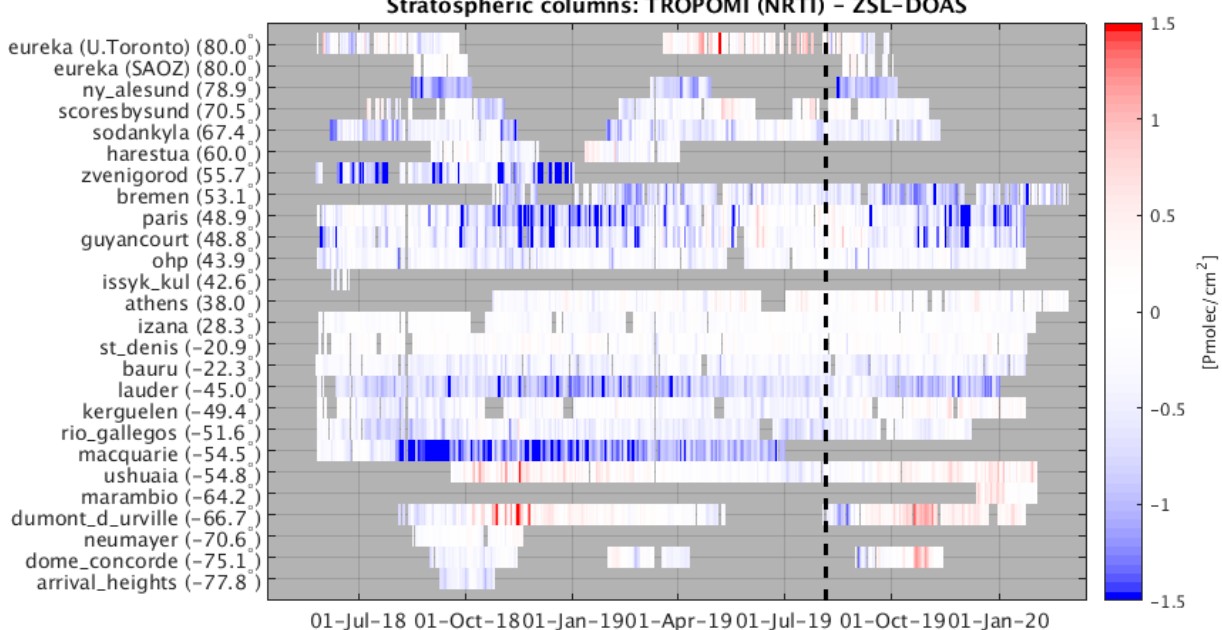

**Figure 5.** Difference between the S5p TROPOMI and NDACC ZSL-DOAS $NO_2$ stratospheric column data as a function of time, after photo-chemical adjustment of the ZSL-DOAS sunset data to the S5p SZA. Stations are ordered by increasing latitude (South at the bottom). The dashed vertical line on August 6, 2019, represents the reduction in S5p ground pixel size from $7.0 \times 3.5 km^2$ to $5.5 \times 3.5 km^2$.

the TROPOMI measurement. This evaluation does not reveal any variation of the bias much larger than 0.4 Pmolec/cm$^2$ over
the range of those influence quantities.

### 3.4   PGN measurements at high-altitude stations

Three of the PGN direct-sun instruments (see Sect. 5) are located near the summit of a volcanic peak: Altzomoni (3985m a.m.s.l) in the State of Mexico, Izaña (2360m a.m.s.l.) on Mount Teide on the island of Tenerife, and Mauna Loa (4169m a.m.s.l.) on the island of Hawaii. At these high-altitude sites the total column measured by the ground-based direct-sun in-
strument misses most of the tropospheric (potentially polluted) part and as such becomes representative of the TROPOMI stratospheric column. These sites have therefore been added to Fig. 6, illustrating that these comparisons based on direct-sun data yield similar conclusions as those based on zenith-sky data, that is, a minor negative median difference of the order of -0.2 Pmolec/cm$^2$. It must be noted that the PGN data are processed using cross sections at a single temperature, representative for the troposphere (254 K). This leads to columns which are about 10% larger than if they had been processed with cross



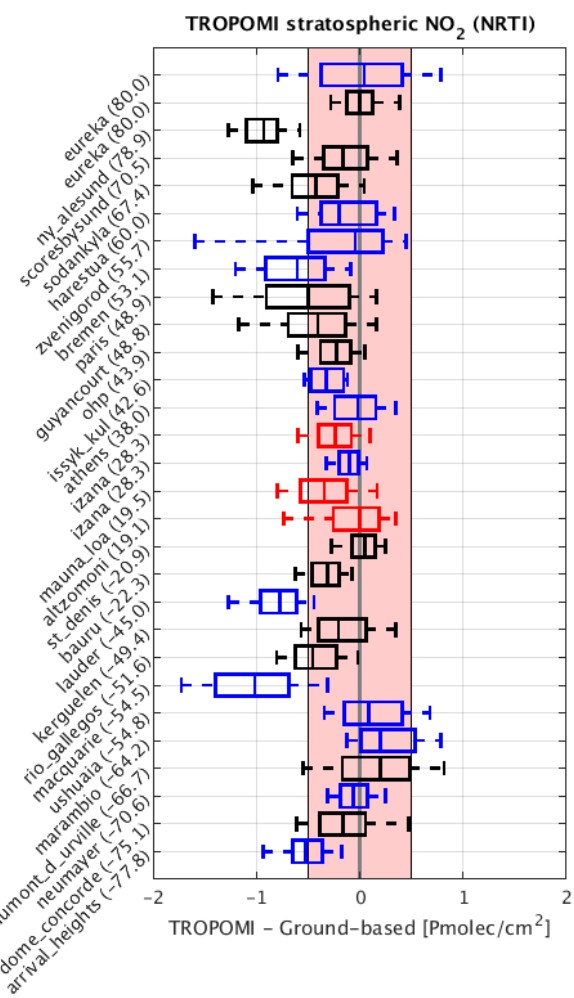

**Figure 6.** Box-and-whisker plots summarizing from pole to pole the bias and spread of the difference between S5p TROPOMI NRTI and NDACC ZSL-DOAS $NO_2$ stratospheric columns (SAOZ data in black, other ZSL-DOAS in blue, and PGN in red). The median difference is represented by a vertical solid line inside the box, which marks the 25 and 75% quantiles. The whiskers cover the 9-91% range of the differences. The shaded area represents the mission requirement of 0.5 Pmolec/cm$^2$ for the uncertainty. Values between brackets in the labels denote the latitude of the station.

sections for 220 K. Future processing of the PGN data will address this, and it is expected that this will mostly remove the apparent negative bias for TROPOMI (but lead to a slight inconsistency with the ZSL-DOAS results).



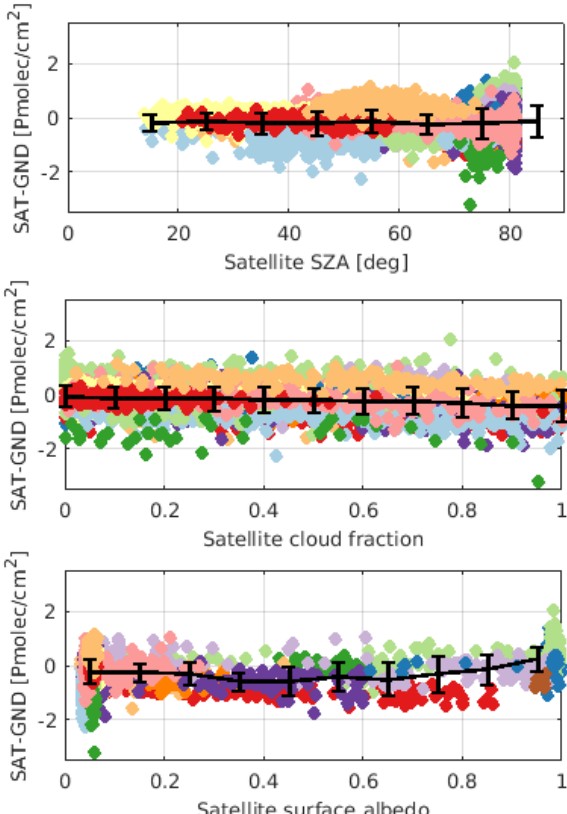

**Figure 7.** Dependence of the difference between TROPOMI OFFL and ground-based ZSL-DOAS stratospheric $NO_2$ column data on the satellite solar zenith angle (SZA), satellite cloud fraction, and satellite surface albedo, including a median and IP68/2 spread per bin (bin widths of 10 degrees in SZA, 0.1 in CF, and 0.1 in surface albedo).

## 4   Tropospheric column validation

### 4.1   MAX-DOAS data

Satellite tropospheric $NO_2$ column data are compared clasically to correlative measurements acquired by MultiAxis-
Differential Optical Absorption Spectroscopy (MAX-DOAS) instruments (Hönninger and Platt, 2002; Honninger et al., 2004; Sinreich et al., 2005). MAX-DOAS instruments measure from sunrise to sunset the UV-visible radiance scattered in several directions and elevation angles, from which the tropospheric VCD and/or the lowest part of the tropospheric $NO_2$ profile (usually up to 3km altitude, and up to 10km at best) can be retrieved through different techniques (see e.g. Clémer et al., 2010; Hendrick et al., 2014; Friedrich et al., 2019; Bösch et al., 2018; Irie et al., 2008, 2011; Vlemmix et al., 2010; Wagner et al.,





2011; Beirle et al., 2019), with between 1 and 3 degrees of freedom. Their horizontal spatial representativeness varies with the aerosol load and the spectral region of the retrieval, from a few km to tens of km (Irie et al., 2011; Wagner et al., 2011; Wang et al., 2014). The total uncertainty estimate on the $NO_2$ tropospheric VCD is of the order of 7-17% in polluted conditions, including both random (around 3 to 10% depending on the instrument) and systematic (11 to 14%) contributions (Irie et al., 2008; Wagner et al., 2011; Hendrick et al., 2014; Kanaya et al., 2014).

MAX-DOAS data have been used extensively for tropospheric $NO_2$ satellite validation, for instance for Aura OMI and MetOp GOME-2 (e.g. by Celarier et al., 2008; Irie et al., 2012; Lin et al., 2014; Kanaya et al., 2014; Wang et al., 2017; Drosoglou et al., 2018; Liu et al., 2019a; Compernolle et al., 2020b; Pinardi et al., 2020), as well as for the evaluation of modelling results (Vlemmix et al., 2015; Blechschmidt et al., 2020).

Data are collected either through ESA's Atmospheric Validation Data Centre (EVDC, https://evdc.esa.int/) or by direct delivery from the instrument Principal Investigators (e.g. within the S5PVT NIDFORVAL AO project). Currently, 19 MAX-DOAS stations have contributed correlative data in the TROPOMI measurement period from April 2018 to February 2020. Detailed information about the stations and instruments is provided in A2. A few contributing sites measure in several geometries (e.g., Xianghe measure in both MAX-DOAS and direct sun mode, Bremen and Athens both report MAX-DOAS and zenith-sky measurements) or have multiple instruments (e.g., Cabauw and UNAM stations host both MAX-DOAS and Pandora instruments). This allows detailed (sub)column consistency-checks and in-depth analysis of the site peculiarities, out of the scope of the present overview paper.

### 4.2 Co-location and harmonization

TROPOMI data is filtered following the qa_value$> 0.75$ rule as recommended in the associated PRF (see Sect. 2). Then for each day, the pixel over the site is selected. MAX-DOAS data series are temporally interpolated at the TROPOMI overpass time (only if data within $\pm$1h exist) and daily comparisons are performed. This short temporal window avoids the need for a photochemical cycle adjustment. Details on the comparison approach are described in Pinardi et al. (2020) for the validation of OMI and GOME-2 $NO_2$ column data and in Compernolle et al. (2020b) for the validation of the OMI QA4ECV $NO_2$ Climate Data Record.

### 4.3 Comparison results

An illustration of the daily comparisons between TROPOMI and ground-based MAX-DOAS measurements between May 2018 and end of January 2020, is presented in Fig. 8 for the Uccle station (Brussels, B, with moderate pollution levels). The two datasets have a correlation coefficient of 0.75 and a regression slope and intercept of 0.47 and 1.0 Pmolec/cm$^2$ respectively. The (median and mean) difference of about -2.3 to -3.1 Pmolec/cm$^2$ corresponds to a median relative difference of about -30%.

Results for the entire MAX-DOAS network are presented in Fig. 9. This figure reveals mostly (but not only) negative differences with a fairly significant variability but no clear seasonal features. No impact of the TROPOMI ground pixel size change on August 6, 2019, is observed.

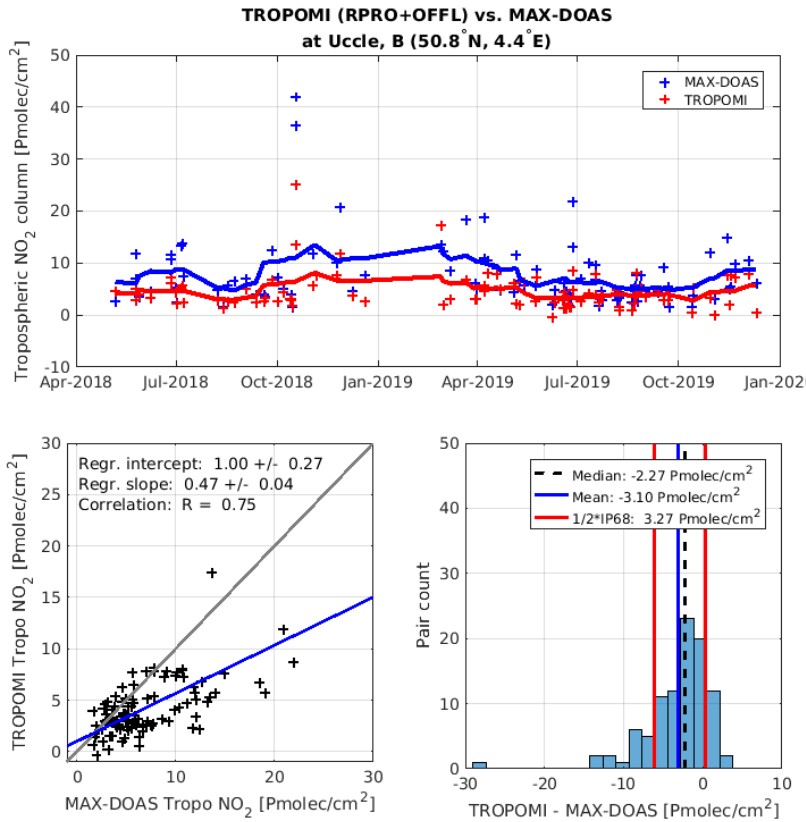

**Figure 8.** Same as Fig. 4, but now for the S5p OFFL tropospheric $NO_2$ column data co-located with ground-based MAX-DOAS measurements performed by BIRA-IASB at the NDACC mid-latitude station of Uccle in Brussels (Belgium).

Box-whisker plots for the whole network are shown in Fig. 10, with corresponding numeric values listed in A2. Based on measurements from these 19 MAX-DOAS stations, three different regimes can be identified:

(i) Small tropospheric $NO_2$ column values (median values below 2 Pmolec/cm$^2$), e.g. at the Fukue and Phimai stations, lead
to small differences. Typically, these stations show a small median biase (<0.5 Pmolec/cm$^2$), but these can still correspond to up to a -27% relative bias. The dispersion (IP68/2) of the difference is smaller than 1 Pmolec/cm$^2$.

(ii) More polluted sites (median tropospheric columns from 3 to 14 Pmolec/cm$^2$) experience a clear negative bias. The median difference ranges between -1 and -5 Pmolec/cm$^2$, i.e. between -15% (Chiba) and -56% (Pantnagar). This underestimation is similar to the one identified in the validation of Aura OMI and MetOp GOME-2 tropospheric $NO_2$ data by Compernolle
et al. (2020b) and Pinardi et al. (2020). The dispersion (IP68/2) of the difference ranges from ~2 to ~6 Pmolec/cm$^2$, roughly increasing with increasing tropospheric $NO_2$ median VCD.

(iii) Extremely polluted sites report larger differences. This is the case e.g. at the Mexican UNAM sites (UNAM and Vallejo in/close to Mexico city, and Cuautitlan in a more remote part of the state of Mexico), with median tropospheric columns larger

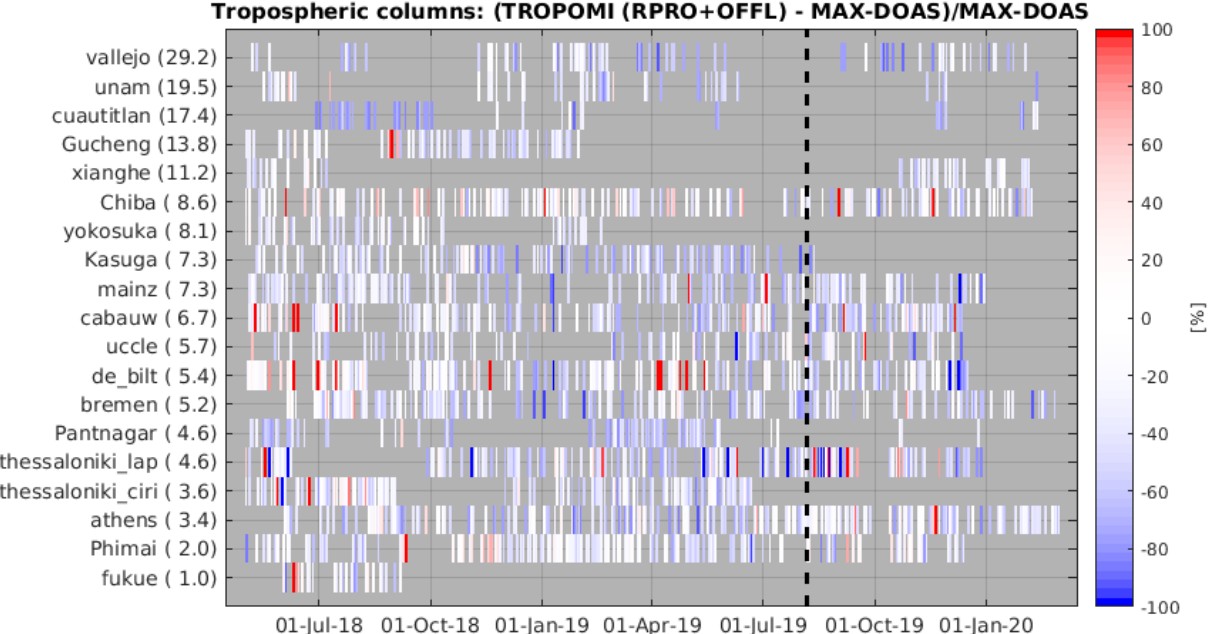

**Figure 9.** Percent relative difference between the S5p TROPOMI and MAX-DOAS $NO_2$ tropospheric column data as a function of time. Stations are ordered by median $NO_2$ tropospheric column (lowest median value at the bottom). The dashed vertical line on August 6, 2019, represents the reduction in S5p ground pixel size from $7.0 \times 3.5 km^2$ to $5.5 \times 3.5 km^2$.

than 15 Pmolec/$cm^2$. These stations experience larger differences (>10 Pmolec/$cm^2$, i.e., from -37 to -74%). The dispersion
(IP68/2) of the difference is also quite large, between 4 and ~12 Pmolec/$cm^2$. Results at these sites need deeper analysis.

The overall bias (median of all station median differences) is -2.4 Pmolec/$cm^2$, i.e. -37%. The median dispersion is 3.5 Pmolec/$cm^2$ while the site-to-site dispersion (IP68/2 over all site medians) is 2.8 Pmolec/$cm^2$. Note that these numbers over all sites are close to the numbers found for the polluted (Athens to Gucheng) sites. These results are within the mission requirement of a maximum bias of 50%, but exceed the uncertainty requirement of maximum 0.7 Pmolec/$cm^2$, which is however
reached for the clean sites ensemble.

## 5 Total column validation

### 5.1 PGN/Pandora data

The Pandonia Global Network (PGN) delivers direct-sun total column and multi-axis tropospheric column observations of several trace gases including $NO_2$ from a network of ground-based standardized Pandora sunphotometers in an automated





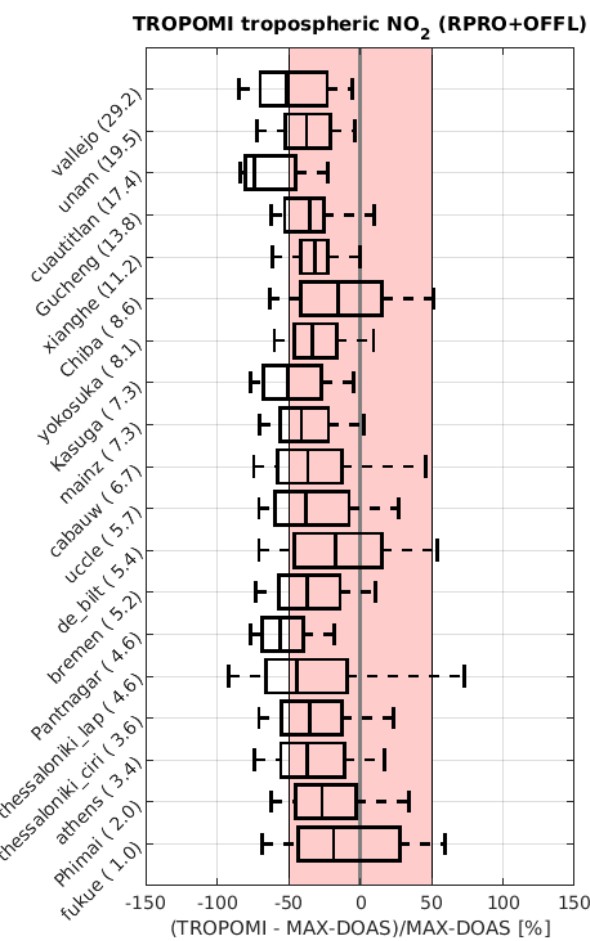

**Figure 10.** Same as Fig. 6, but now for the difference between S5p TROPOMI OFFL and MAX-DOAS NO$_2$ tropospheric columns, and with ordered as a function of the median ground-based tropospheric column (largest median VCD values on top). The line represents the median difference. Box bounds represent 25 and 75 percentile, while whiskers indicate the 9 and 91 percentiles. The shaded area corresponds to the mission requirement of maximum 50% for the bias.

way. In this work, only direct-sun observations are used. These have a random error uncertainty of about 0.27Pmolec/cm$^2$ and a systematic error uncertainty of 2.7Pmolec/cm$^2$ (Herman et al., 2009). Studies at US and Korean sites during DISCOVER-AQ campaign found a good agreement of Pandora instruments with aircraft in-situ measurements (within 20 percent on average Choi et al., 2019), although larger differences are observed for individual sites (Nowlan et al., 2018).





Pandora data have been used before to validate satellite $NO_2$ measurements from Aura OMI (Herman et al., 2009; Tzortziou
et al., 2014; Kollonige et al., 2018; Choi et al., 2019; Judd et al., 2019; Griffin et al., 2019; Herman et al., 2019; Pinardi et al.,
2020) and TROPOMI (Griffin et al., 2019; Ialongo et al., 2020; Zhao et al., 2019).

For the current work, 25 sites have contributed Pandora data, collected either from the ESA Atmospheric Validation Data
Centre (EVDC) (https://evdc.esa.int/) or from the PGN data archive (https://pandonia-global-network.org/). Only data files
from a recent quality upgrade (processor version 1.7, retrieval version nvs1, with file version 004 and 005; see https://www.
pandonia-global-network.org/home/documents/release-notes/) were used, with 005 files (consolidated data) having precedence
over 004 files (rapid delivery data). The most important change with the previous data release is a more stringent quality
filtering. Seventeen sites have provided measurement data newer than 3 months.

Except at low sun elevation, the footprint of these direct-sun measurements is much smaller than a TROPOMI pixel. There-
fore, - as it is the case with MAX-DOAS - a significant horizontal smoothing difference error can be expected in the TROPOMI-
Pandora comparison, especially in the case of tropospheric $NO_2$ gradients and when tropospheric $NO_2$ is the largest contributor
to the total column.

Three Pandora instruments (Altzomoni, Izaña, Mauna Loa) are located near the summit of a volcanic peak and are therefore
not sensitive to the lower-lying tropospheric $NO_2$. In this work, their observations are compared to the TROPOMI stratospheric
$NO_2$ data (see Sect. 3).

## 5.2 Filtering, co-location and harmonization

As was done for the tropospheric column validation in Sect. 4, only S5p pixels with qa_value at least 0.75 are retained. The
so-called summed product is used, i.e. the total column computed as the stratospheric plus the tropospheric column values. This
summed column differs from the total column product. Only Pandonia measurements with the highest quality label (0 and 10)
are used. The average column value within a 1-hour time interval, centered on the S5p overpass time, is used. As the $NO/NO_2$
ratio varies only slowly around the afternoon solar local time of the TROPOMI overpass, this small temporal window ensures
no model-based adjustment is required. A 30-minute time interval was tested as well, but this did not change significantly the
results. Moreover, only TROPOMI pixels containing the station were considered.

## 5.3 Comparison results

An example of a time series of co-located TROPOMI and PGN total column measurements, and their difference, is shown in
Fig. 11.

Results for the entire PGN network are presented in Fig. 12. This figure reveals that the difference, even in relative units,
depends strongly on the total $NO_2$ column, with low (or slightly positive) biases at low columns, and markedly negative biases
at high columns. No impact is observed for the TROPOMI ground pixel size switch of August 6, 2019.

Statistical estimators of the comparison results across the network are visualized in Fig. 13 and presented in tabular form in
A3. One can distinguish roughly two different regimes.



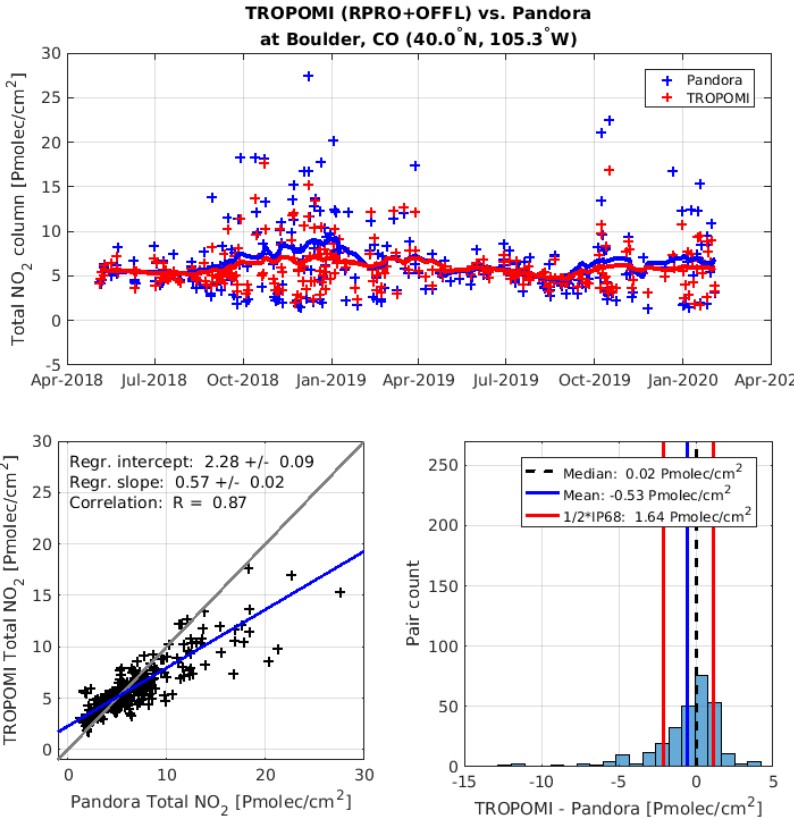

**Figure 11.** Same as Fig. 4 and Fig. 8, but now for the S5p OFFL total $NO_2$ column data co-located with ground-based Pandora measurements obtained at the PGN mid-latitude station of Boulder, Colorado.

(i) PGN median total column value between 3 (Alice Springs) and 6 Pmolec/cm$^2$ (New Brunswick). The absolute bias (median difference) is within $\pm0.2$ Pmolec/cm$^2$ in most cases (up to +0.5 Pmolec/cm$^2$ at Egbert and Helsinki) while the median relative difference is within 5% in most cases (up to ~ 10% at Alice Springs, Egbert, Inoe and Helsinki). Canberra is a deviating case with larger negative bias (-0.9 Pmolec/cm$^2$; -20%) The difference dispersion (IP68/2) roughly increases with

increasing PGN $NO_2$ median VCD, from 0.4-0.6 Pmolec/cm$^2$ at the three cleanest sites, to 1-2 Pmolec/cm$^2$ at the other sites.

(ii) PGN $NO_2$ median total column value between 8 (Buenos Aires) and 19 Pmolec/cm$^2$ (UNAM, Mexico city). A negative bias is observed, ranging from -1 Pmolec/cm$^2$ (-15%) at The Bronx (New York) to -7 Pmolec/cm$^2$ (-50%) at Rome Sapienza. The difference dispersion ranges from ~3 (Buenos Aires) to 5 Pmolec/cm$^2$ (UNAM).

The median relative difference is mostly within (or bordering) the $\pm10\%$ range for the sites with lower $NO_2$ median total

column values (Alice Springs to New Brunswick; Canberra is an exception), while it is negative and mostly outside this range, but still within $\pm50\%$, for the sites with higher $NO_2$ median total column value (Buenos Aires to UNAM).

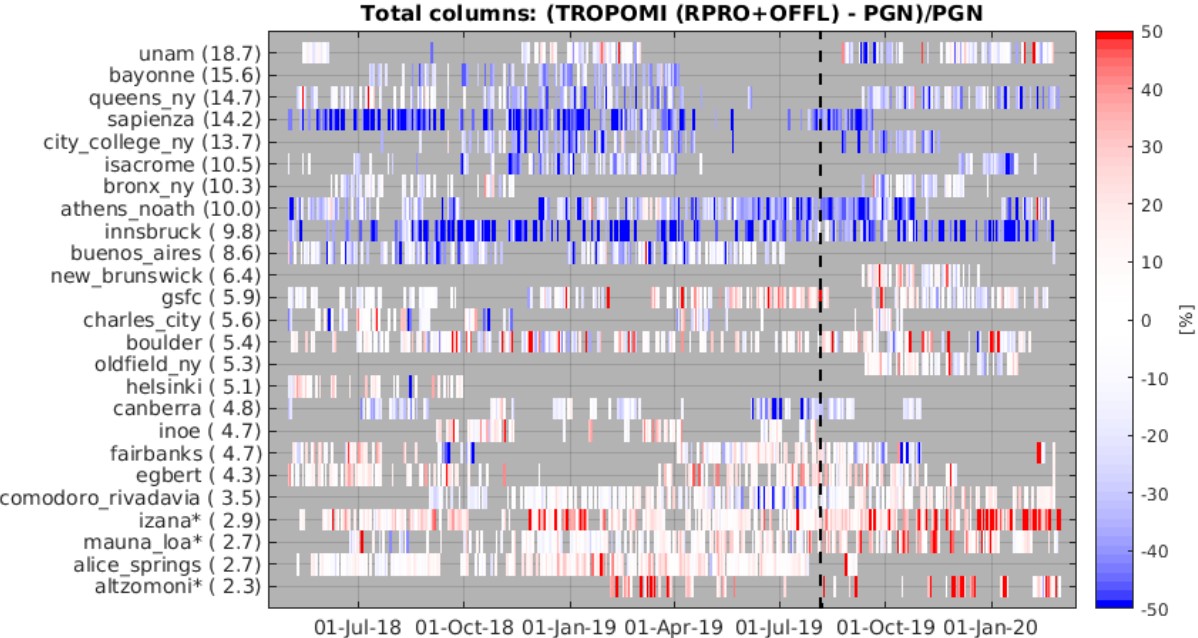

**Figure 12.** Percent relative difference between the S5p TROPOMI and PGN $NO_2$ total column data as a function of time. Stations are ordered by median $NO_2$ total column (lowest median value at the bottom). The dashed vertical line on August 6, 2019, represents the reduction in S5p ground pixel size from $7.0 \times 3.5 \text{km}^2$ to $5.5 \times 3.5 \text{km}^2$. The three mountain-top sites more suited for the validation of only the stratospheric column are marked with an asterisk.

The overall bias over all sites (median over all site medians or site relative medians) is -0.5 Pmolec/cm$^2$ (-7%). The overall dispersion is 1.8 Pmolec/cm$^2$ while the site-to-site dispersion (IP68/2 over all site medians) is 2.2 Pmolec/cm$^2$.

It is however more useful to make the distinction between sites with low $NO_2$ (Alice Springs to New Brunswick) and high

$NO_2$ (Buenos Aires to UNAM). For the low $NO_2$ sites, the overall bias is 0.1 Pmolec/cm$^2$ (2%), the overall dispersion is 1.1 Pmolec/cm$^2$ and the site-to-site dispersion is 0.2 Pmolec/cm$^2$. For the high $NO_2$ sites, the overall bias is -3.6 Pmolec/cm$^2$ (-32%), the overall dispersion is 3.3 Pmolec/cm$^2$ and the site-to-site dispersion is 1.4 Pmolec/cm$^2$.

The slight positive bias at clean sites may be related to the small negative bias observed for the stratospheric columns, but in view of the different uncertainty terms in this validation exercise, it is at the moment impossible to test this in-depth.

**6  Discussion and conclusions**

A cross-networks summary of the median difference and dispersion for the three S5p $NO_2$ (sub)column data is attempted in Table 2. While the difference between the NRTI and OFFL $NO_2$ values can reach up to a few Pmolec/cm$^2$ for individual





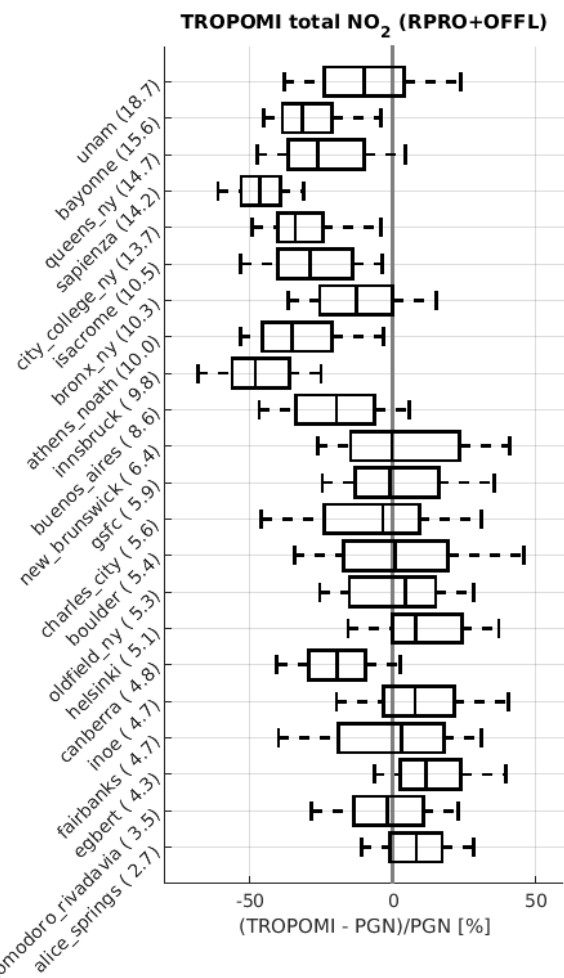

**Figure 13.** Same as Fig. 6 and Fig. 10, but now for the difference between S5p TROPOMI (RPRO+OFFL) and PGN NO$_2$ total columns. Stations are ordered by ground-based total NO$_2$ median value, like in Fig. 10. The median difference is represented by a vertical solid line inside the box, which marks the 25 and 75% quantiles. The whiskers cover the 9-91% range of the differences. The 3 mountain-top PGN instruments used for the validation of the stratospheric columns are not included here, but in Fig. 6.

TROPOMI pixels, the two processing channels do not lead to significantly different validation results, and Table 2 therefore makes no distinction between the two.

For the stratospheric column, the general picture is a slight negative bias of TROPOMI with respect to the NDACC ZSL-DOAS network, of the order of -0.2 Pmolec/cm$^2$ on an average, with some station-to-station inhomogeneities and with larger differences in the highly variable conditions of the denoxified polar stratosphere in winter. This bias remains within the S5p





**Table 2.** Cross-networks summary of the validation results: bias (median) and dispersion (IP68/2) of the difference w.r.t the ground-based correlative measurements (median value over the stations).

|  | Bias | Dispersion |
|---|---|---|
| **Stratosphere** | -0.2 Pmolec cm$^{-2}$; -9% | 0.3 Pmolec/cm$^2$ |
| **Troposphere** | | |
| low NO$_2$ | -0.3 Pmolec/cm$^2$; -23% | 0.7 Pmolec/cm$^2$ |
| high NO$_2$ | -2 Pmolec/cm$^2$; -37% | 3.4 Pmolec/cm$^2$ |
| extreme NO$_2$ | -12 Pmolec/cm$^2$; -51% | 7 Pmolec/cm$^2$ |
| **Total column** | | |
| low NO$_2$ | 0.1 Pmolec/cm$^2$ ; 2% | 1 Pmolec/cm$^2$ |
| high NO$_2$ | -3.6 Pmolec/cm$^2$ ; -30% | 3 Pmolec/cm$^2$ |

mission requirements and is similar to the conclusions derived for similar satellite data from other sounders (e.g., Compernolle et al., 2020b). In view of the sources of systematic uncertainties in the different components of the comparison (satellite data,

reference data, photochemical cycle adjustment, irreducible mismatch errors), this bias is entirely within expectations.

For the tropospheric and total columns, averaging results over the networks with the hope to obtain a meaningful global estimate is of limited use as the results depend strongly on the amount of tropospheric NO$_2$. Overall, mission requirements in terms of bias are mostly met, the only exception being the tropospheric columns at extremely polluted sites, which have a bias on the threshold of 50%. Nevertheless, it is clear that large negative median differences are observed across all sites

experiencing significant tropospheric pollution. The dispersion of the difference is well outside of the mission requirements formulated for the tropospheric column data. Nevertheless, these results are consistent with those obtained with completely different validation techniques, such as explored by Lorente et al. (2019) over Paris (using ground-based and Eiffel Tower NO$_2$ concentrations and a climatology of observed column-to-surface ratios). Many factors play a role in this apparent disagreement between TROPOMI and the ground-based networks, that cannot all be attributed to the S5p data.

First: Local horizontal and vertical variations of the NO$_2$ field can explain (part of) such discrepancies, as illustrated in Chen et al. (2009); Pinardi et al. (2020); Compernolle et al. (2020b); Dimitropoulou et al. (2020). While the MAX-DOAS picks up small local enhancements, the much larger satellite pixel provides a smoothed perception of the field. This generally leads to under-estimation in urban conditions while having better agreement in remote locations (Celarier et al., 2008; Kanaya et al., 2014; Pinardi et al., 2020), as it is the case in the current study. Dimitropoulou et al. (2020) showed specific improvements

of the S5p NO$_2$ comparison results in the case of the Uccle MAX-DOAS when making use of the multiple azimuthal scan mode and when improving the S5p selection criteria to pixels along the MAX-DOAS field of view direction and within the effective sensitivity length. Large inhomogeneities around MAX-DOAS sites were also shown by (Wang et al., 2014; Ortega et al., 2015; Gratsea et al., 2016; Peters et al., 2019; Schreier et al., 2020). When taking part of these inhomogeneities into account in validation of other sounders, results had been improved (Brinksma et al., 2008). Judd et al. (2019) also showed





the smoothing of the $NO_2$ field when re-sampling GeoTASO high-resolution airborne measurements to different simulated
satellite pixel sizes.

Second: Vertical sensitivity (and thus averaging kernels) and a priori vertical profiles are known to be different for MAX-
DOAS and nadir UV-visible satellite retrievals (Wang et al., 2017; Liu et al., 2019b; Compernolle et al., 2020b), with MAX-
DOAS measurements sensitive to layers close to the surface and satellite retrievals sensitive mostly to the free troposphere. The
effect of the a-priori vertical profile on the comparison was estimated for TROPOMI by Dimitropoulou et al. (2020) for Uc-
cle, showing an increase by about 55% when recalculating the TROPOMI column with MAX-DOAS daily mean tropospheric
profile. Similarly, Ialongo et al. (2020) and Zhao et al. (2019) show improvement of the agreement between TROPOMI and Pan-
dora total column data for episodes of $NO_2$ enhancement, when replacing the coarse a-priori $NO_2$ profiles with high-resolution
profiles from a high-resolution regional air quality forecast model. Explicit aerosol corrections in the satellite retrievals may
further improve the agreement (Liu et al., 2020).

Third: The treatment of cloud properties can have a significant effect on the retrieval of the TROPOMI $NO_2$ tropospheric
VCD. Eskes et al. (2020) discuss the comparison with OMI $NO_2$ tropospheric column retrievals and show that on an average
TROPOMI is lower than OMI by -10% to -12% over Europe, North America and India, and up to -22% over China. This
difference is mainly attributed to the different cloud data product used in the $NO_2$ retrieval: FRESCO-S derives the cloud
top pressure from TROPOMI radiances in the near-infrared $O_2$-A band, while for OMI the cloud top pressure is retrieved
from the $O_2$-$O_2$ band in the UV-Visible. Preliminary validation results (Compernolle et al., 2020a, and H. Eskes, private
communication) indicate that FRESCO-S is biased high in pressure, especially at altitudes close to the surface. A new version
of FRESCO-S with an adapted wavelength window has been implemented and seems to remove most of the 10-22% bias with
OMI in polluted regions.

Fourth: Although this work, Compernolle et al. (2020b), and Pinardi et al. (2020) all show a generally good coherence
of the validation results among the MAX-DOAS instruments across the network and also among MAX-DOAS and Pandora
instruments, network homogenization remains an important challenge to focus on to improve the accuracy of future satel-
lite validations. Inter-comparison campaigns, such as the CINDI-1 and -2 (Piters et al., 2012; Kreher et al., 2019), in-depth
intercomparison studies of the retrieval methods (Frieß et al., 2019; Tirpitz et al., 2020; Peters et al., 2019), and dedicated
projects aiming at the harmonization of the processing and of the associated metadata (such as the FRM4DOAS project of
ESA's Fiducial Reference Measurements programme) are an important way to achieve this.

Regarding the mutual consistency of MAX-DOAS and PGN based validation results: while it may appear that, at low
column values, PGN base comparisons indicate a smaller bias than the MAX-DOAS comparisons, one must not forget that
PGN measures the total column: at stations with a lower total column value, the stratospheric contribution is relatively more
important. The better agreement here is therefore consistent with the good agreement found for the TROPOMI stratospheric
$NO_2$ column vs. ZSL-DOAS and also vs. PGN at pristine mountain sites (Section 3). For sites characterised by a higher total
$NO_2$ column, the tropospheric contribution becomes more important, and some of the same effects that make satellite-to-
MAX-DOAS comparisons difficult, such as smoothing difference error, lower sensitivity of the satellite close to the surface,
and approximate S5p a-priori profile, come into play as well.





In conclusion, the first two years of Copernicus S5p TROPOMI $NO_2$ column data produced both with the NRTI and OFFL versions 01.0x.xx of the operational processors, do meet mission requirements for the bias, and to some extent and with precaution for the uncertainty (dispersion). The different data products available publicly through the Copernicus system are mutually consistent, in good geophysical and quantitative agreement with ground-based correlative data of documented quality, and can be used for a variety of applications, on the condition that the features and limitations exposed here are taken into

proper consideration, and that the S5p data are filtered and used according to the recommendations provided in the official Product Readme File (PRF) and associated documentation, also available publicly. Ground-based validation activities relying on the correlative measurements contributed by the NDACC ZSL-DOAS, MAX-DOAS and PGN global monitoring networks, have progressed significantly in recent years and have demonstrated their capacity, but also their current limitations in an operational context such as the Copernicus programme. Room does exist for further improvement of both the satellite and

ground-based data sets, as well as the intercomparison methodology and its associated error budget. Beyond the methodology advances published here and in aforementioned papers, special effort is needed to understand fully and ever reduce comparison mismatch errors, which so far make difficult the accurate validation of S5p data uncertainty bars. Several updates of the calibration of TROPOMI spectra and of the TROPOMI $NO_2$ data retrieval processors are already in development and in implementation. Upcoming data versions should be validated with the same system as used in the current paper, allowing the

necessary independent assessment of the S5p data product evolution.

**Appendix A: Ground networks**

**A1 The NDACC ZSL-DOAS network**





**Table A1.** ZSL-DOAS hosting stations, ordered by latitude, that are contributing to the stratospheric $NO_2$ column validation. Several measures of the agreement between TROPOMI and the ground-based data are also provided. The bias over all stations (median over all station median differences) is $-0.23\ \text{Pmoleccm}^{-2}$ while the overall dispersion (median over all $1/2\text{IP68}$) is $0.31\ \text{Pmoleccm}^{-2}$ and the inter-station dispersion ($1/2\text{IP68}$ over all station medians) is $0.30\ \text{Pmoleccm}^{-2}$.

| Station | Latitude [deg] | Longitude [deg] | Altitude [m] a.m.s.l. | Institute | Processing | Median diff. [Pmolec/cm²] | Spread (IP68/2) [Pmolec/cm²] | R |
|---|---|---|---|---|---|---|---|---|
| Eureka | 80.05 | -86.42 | 610 | U. Toronto | NDACC | 0.04 = 1% | 0.60 | 0.89 |
| Eureka | 80.05 | -85.42 | 610 | LATMOS-CNRS + U. Toronto | LATMOS_RT | -0.00 = 0% | 0.20 | 0.97 |
| Ny-Ålesund | 78.92 | 11.93 | 10 | NILU | LATMOS_RT | -0.93 = -26% | 0.24 | 0.97 |
| Scoresbysund | 70.48 | -21.95 | 67 | LATMOS-CNRS + DMI | LATMOS_RT | -0.16 = -5% | 0.32 | 0.98 |
| Sodankylä | 67.37 | 26.63 | 179 | LATMOS-CNRS + FMI | LATMOS_RT | -0.42 = -12% | 0.37 | 0.97 |
| Harestua | 60.00 | 10.75 | 596 | BIRA-IASB | NDACC | -0.19 = -8% | 0.35 | 0.82 |
| Zvenigorod | 55.69 | 36.77 | 220 | IAP, RAS | NDACC | -0.64 = -20% | 0.73 | NaN |
| Bremen | 53.10 | 8.85 | 27 | IUP Bremen | NDACC | -0.60 = -19% | 0.40 | 0.91 |
| Paris | 48.85 | 2.35 | 63 | LATMOS-CNRS | LATMOS_RT | -0.50 = -16% | 0.56 | 0.59 |
| Guyancourt | 48.78 | 2.03 | 160 | LATMOS-CNRS | LATMOS_RT | -0.40 = -13% | 0.45 | 0.71 |
| Haute Provence (O.H.P.) | 43.94 | 5.71 | 650 | LATMOS-CNRS | LATMOS_RT | -0.23 = -8% | 0.23 | 0.94 |
| Issyk-Kul | 42.62 | 76.99 | 1640 | KNU | NDACC | -0.33 = -9% | 0.19 | 0.48 |
| Athens | 38.05 | 23.86 | 527 | IUP Bremen + NOA | NDACC | -0.02 = -1% | 0.28 | 0.89 |
| Izaña | 28.31 | -16.50 | 2367 | INTA | NDACC | -0.10 = -4% | 0.14 | 0.95 |
| Saint-Denis | -20.90 | 55.48 | 110 | LATMOS-CNRS + LACy | LATMOS_RT | 0.05 = 2% | 0.18 | 0.80 |
| Bauru | -22.35 | -49.03 | 640 | LATMOS-CNRS + UNESP | LATMOS_RT | -0.31 = -12% | 0.19 | 0.80 |
| Lauder | -45.04 | 169.68 | 370 | NIWA | NDACC | -0.77 = -23% | 0.28 | 0.92 |
| Kerguelen | -49.35 | 70.26 | 36 | LATMOS-CNRS | LATMOS_RT | -0.21 = -7% | 0.34 | 0.94 |
| Rio Gallegos | -51.60 | -69.32 | 15 | LATMOS-CNRS | LATMOS_RT | -0.45 = -16% | 0.28 | 0.95 |
| Macquarie | -54.50 | 158.94 | 6 | NIWA | NDACC | -1.01 = -27% | 0.48 | 0.93 |
| Ushuaïa | -54.82 | -68.32 | 7 | INTA | NDACC | 0.09 = 4% | 0.40 | 0.95 |
| Marambio | -64.23 | -56.72 | 198 | INTA | NDACC | 0.20 = 4% | 0.39 | 0.50 |
| Dumont d'Urville | -66.67 | 140.02 | 45 | LATMOS-CNRS | LATMOS_RT | 0.20 = 5% | 0.50 | 0.95 |
| Neumayer | -70.63 | -8.25 | 43 | U. Heidelberg | NDACC | -0.06 = -5% | 0.21 | 0.95 |
| Dome Concorde | -75.10 | 123.31 | 3250 | LATMOS-CNRS | LATMOS_RT | -0.16 = -6% | 0.38 | 0.95 |
| Arrival Heights | -77.83 | 166.66 | 184 | NIWA | NDACC | -0.52 = -26% | 0.25 | 0.90 |





## A2 The MAX-DOAS network

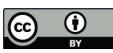



Atmospheric Measurement Techniques Discussions — Open Access — EGU

**Table A2.** MAX-DOAS hosting stations, ordered by increasing median tropospheric column ($VCDgb$, lowest at the bottom), that are contributing to the tropospheric $NO_2$ column validation. More details on the QA4ECV data sets can be found at http://www.qa4ecv.eu/ecvs. Several measures of the agreement between TROPOMI and the ground-based data are also provided. Biases and comparison spreads vary strongly between stations, mainly as a function of the nature of the site (clean or polluted). When calculating these numbers for the three regimes (clean, polluted, extreme), the median biases are: -0.3, -2 and -12 $Pmolec/cm^2$ (-23%, -37% and -51%) respectively, with median dispersions of: 0.7, 3.4 and 7 $Pmolec/cm^2$. Note that the median values for the high tropospheric columns (Athens to Xianghe) are almost the same as the statistics found for the whole network. The site-to-site bias dispersion is 0.2, 1.2 and 3.3 $Pmolec/cm^2$ for each regime.

| Station | Latitude [deg] | Longitude [deg] | Altitude [m a.m.s.l.] | Institute | Reference | median(VCDgb) [Pmolec/cm²] | Median diff. [Pmolec/cm²; %] | Spread (IP68/2) [Pmolec/cm²] | R |
|---|---|---|---|---|---|---|---|---|---|
| Vallejo | 19.48 | -99.15 | 2255 | UNAM | Arellano et al. (2016); Friedrich et al. (2019) | 29 | -14; -51.3% | 12 | 0.40 |
| UNAM | 19.33 | -99.18 | 2280 | UNAM | Arellano et al. (2016); Friedrich et al. (2019) | 19 | -7.8; -37.3% | 7 | 0.84 |
| Cuautitlan | 19.72 | -99.20 | 2263 | UNAM | Arellano et al. (2016); Friedrich et al. (2019) | 17 | -12; -73.8% | 4.3 | 0.70 |
| Gucheng | 39.15 | 115.73 | 13.4 | USTC | Xing et al. (2017, 2020) | 14 | -5.4; -35.3% | 6.5 | 0.86 |
| Xianghe | 39.75 | 116.96 | 95 | BIRA-IASB | Hendrick et al. (2014) | 11 | -3.9; -31.7% | 5.7 | 0.83 |
| Chiba | 35.60 | 140.10 | 21 | ChibaU | Irie et al. (2011, 2012, 2015) | 8.6 | -1; -15% | 6.3 | 0.79 |
| Yokosuka | 35.32 | 139.65 | 10 | JAMSTEC | Kanaya et al. (2014) | 8.1 | -2.4; -33% | 3.7 | 0.85 |
| Kasuga | 33.52 | 130.48 | 28 | ChibaU | Irie et al. (2011, 2012, 2015) | 7.3 | -3.1; -50.4% | 4 | 0.46 |
| Mainz | 49.99 | 8.23 | 150 | MPIC | QA4ECV dataset | 7.3 | -3.3; -41% | 3.3 | 0.75 |
| Cabauw | 51.97 | 4.93 | 3 | KNMI | Vlemmix et al. (2010) | 6.7 | -2.5; -36.5% | 3.5 | 0.40 |
| Uccle | 50.80 | 4.36 | 120 | BIRA-IASB | Gielen et al. (2014) | 5.7 | -2.3; -37% | 3.3 | 0.75 |
| De Bilt | 52.10 | 5.18 | 20 | KNMI | Vlemmix et al. (2010) | 5.4 | -0.95; -16.8% | 2.8 | 0.64 |
| Bremen | 53.10 | 8.85 | 27 | IUPB | QA4ECV dataset | 5.2 | -2.1; -37% | 2.3 | 0.59 |
| Pantnagar | 29.03 | 79.47 | 237 | ChibaU | Hoque et al. (2018b) | 4.6 | -2.6; -56% | 1.6 | 0.33 |
| Thessaloniki_lap | 40.63 | 22.96 | 60 | AUTH | Drosoglou et al. (2017), QA4ECV dataset | 4.6 | -1.5; -43.8% | 4.1 | 0.69 |
| Thessaloniki_ciri | 40.56 | 22.99 | 70 | AUTH | Drosoglou et al. (2017), QA4ECV dataset | 3.6 | -1.3; -34.9% | 2 | 0.73 |
| Athens | 38.05 | 23.86 | 527 | IUPB | QA4ECV dataset | 3.4 | -1.1; -36.7% | 3 | 0.66 |
| Phimai | 15.18 | 102.56 | 212 | ChibaU | Hoque et al. (2018a) | 2 | -0.5; -26.6% | 0.7 | 0.47 |
| Fukue | 32.75 | 128.68 | 80 | JAMSTEC | Kanaya et al. (2014) | 0.95 | -0.18; -18.5% | 0.6 | 0.01 |





## A3 The Pandonia Global Network

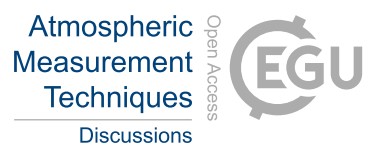
**Table A3.** PGN stations, ordered by median PGN $NO_2$ column value, that are contributing to the total $NO_2$ validation. Mountain-top stations (not sensitive to lower lying tropospheric $NO_2$ are marked with an asterisk. In the last row, we indicated where the data can be obtained (EVDC or directly from the PGN website). Note that only PGN data from a recent quality upgrade (with file version 004 or 005, where 005 has precedence) was used. The bias over all stations (median over all station medians) is -0.5 $Pmoleccm^{-2}$ (-7%) while the overall dispersion (median over all $1/2IP68$) is 1.8 $Pmoleccm^{-2}$ and the inter-station dispersion ($1/2IP68$ over all station medians) is 2.2 $Pmoleccm^{-2}$. Considering the low $NO_2$ stations (Alice Springs to New Brunswick) only, the bias is 0.1 $Pmoleccm^{-2}$ (2%), the overall dispersion is 1.1 $Pmoleccm^{-2}$ and the inter-station dispersion is 0.2 $Pmoleccm^{-2}$. For the high $NO_2$ stations (Buenos Aires to UNAM), the bias is -3.6 $Pmoleccm^{-2}$ (-30%), the overall dispersion is 3.3 $Pmoleccm^{-2}$ and the inter-station dispersion is 1.4 $Pmoleccm^{-2}$. Note that the mountain-top stations are not used in the calculation of these overall statistics.

| Station code | Full name | Lat [°] | Lon [°] | Alt [m] | PGN med(VCD) [$Pmoleccm^{-2}$] | med(diff) ;med(reldiff) [$Pmoleccm^{-2}$] | 1/2IP68(diff) [$Pmoleccm^{-2}$] | R | archive |
|---|---|---|---|---|---|---|---|---|---|
| unam | National Autonomous University of Mexico | 19.33 | -99.18 | 2280 | 18.7 | -2.1;-10% | 4.6 | 0.87 | both |
| Bayonne | Bayonne | 40.67 | -74.13 | 3 | 15.6 | -4.3;-31% | 3.2 | 0.88 | EVDC |
| queens_ny | New York Queens College | 40.74 | -73.82 | 25 | 14.7 | -3.7;-26% | 3.6 | 0.84 | EVDC |
| sapienza | Rome Sapienza | 41.90 | 12.52 | 75 | 14.2 | -6.6;-46% | 4.0 | 0.81 | EVDC |
| city_college_ny | New York City College | 40.82 | -73.95 | 113 | 13.7 | -4.7;-34% | 3.4 | 0.91 | EVDC |
| isacrome | Rome CNR-ISAC | 41.84 | 12.65 | 117 | 10.5 | -2.7;-29% | 3.2 | 0.85 | both |
| bronx_ny | New York - The Bronx | 40.87 | -73.88 | 31 | 10.3 | -1.0;-13% | 3.3 | 0.90 | both |
| athens_noath | Athens National Observatory | 37.99 | 23.77 | 130 | 10.0 | -3.4;-35% | 2.8 | 0.70 | PGN |
| innsbruck | Innsbruck | 47.26 | 11.39 | 616 | 9.8 | -4.7;-48% | 3.4 | 0.59 | PGN |
| buenos_aires | Buenos Aires | -34.56 | -58.51 | 20 | 8.6 | -1.8;-20% | 2.6 | 0.86 | both |
| new_brunswick | New Brunswick (NJ) | 40.46 | -74.43 | 19 | 6.4 | -0.0;-0% | 1.5 | 0.90 | PGN |
| gsfc | Goddard Space Flight Center | 38.99 | -76.84 | 90 | 5.9 | -0.1;-1% | 1.3 | 0.80 | both |
| charles_city | Charles City (VA) | 37.33 | -77.21 | 6 | 5.6 | -0.2;-3% | 2.0 | 0.44 | both |
| boulder | Boulder | 39.99 | -105.26 | 1660 | 5.4 | 0.0;1% | 1.6 | 0.87 | both |
| oldfield_ny | New York - Old Field | 40.96 | -73.14 | 3 | 5.3 | 0.2;5% | 1.1 | 0.93 | both |
| helsinki | Helsinki | 60.20 | 24.96 | 97 | 5.1 | 0.5;8% | 1.0 | 0.77 | EVDC |
| canberra | Canberra | -35.34 | 149.16 | 600 | 4.8 | -0.9;-19% | 0.9 | 0.64 | EVDC |
| inoe | Magurele | 44.34 | 26.01 | 93 | 4.7 | 0.3;8% | 1.0 | 0.79 | EVDC |
| fairbanks | Fairbanks | 64.86 | -147.85 | 227 | 4.7 | 0.1;3% | 1.4 | 0.43 | EVDC |
| egbert | Egbert | 44.23 | -79.78 | 251 | 4.3 | 0.5;12% | 0.6 | 0.88 | PGN |
| comodoro_rivadavia | Comodoro Rivadavia | -45.78 | -67.45 | 46 | 3.5 | -0.1;-2% | 0.6 | 0.56 | PGN |
| izana* | Izana | 28.31 | -16.50 | 2360 | 2.9 | 0.6;19% | 0.5 | 0.53 | both |
| mauna_loa* | Mauna Loa | 19.48 | -155.60 | 4169 | 2.7 | 0.2;6% | 0.5 | 0.43 | both |
| alice_springs | Alice Springs | -23.76 | 133.88 | 567 | 2.7 | 0.2;8% | 0.4 | 0.61 | EVDC |
| altzomoni* | Altzomoni | 19.12 | -98.66 | 3985 | 2.3 | 0.7;28% | 0.6 | 0.64 | both |



*Author contributions.* TV, SC and GP carried out the global validation analysis. JCL, KUE and MVR contributed input and advise at all stages of the analysis. AMF (EVDC), JG (Multi-TASTE) and SN (MPC VDAF-AVS) pre- and/or post-processed the ground-based and satellite data. HJE, KFB, PFL and JPV developed the TROPOMI NO2 data processor. AR, MVR and TW contributed expertise on satellite NO2 data retrieval. AC, FH, KK, MT, APa, JPP and MVR supervise networks operation and contributed ground-based scientific expertise. AD, LSdM and CZ supervise the Copernicus S5p mission, the S5p MPC and the S5PVT. All other co-authors contributed ground-based data and expertise at ground-based stations. TV, SC, GP and JCL wrote and edited the manuscript. All co-authors revised and commented on the manuscript.

*Competing interests.* The authors declare that they have no conflict of interest.

*Acknowledgements.* Part of the reported work was carried out in the framework of the Copernicus Sentinel-5 Precursor Mission Performance Centre (S5p MPC), contracted by the European Space Agency (ESA/ESRIN, Contract No. 4000117151/16/I-LG) and supported by the Belgian Federal Science Policy Office (BELSPO), the Royal Belgian Institute for Space Aeronomy (BIRA-IASB), the Netherlands Space Office (NSO), and the German Aerospace Centre (DLR). Part of this work was carried out also in the framework of the S5p Validation Team (S5PVT) AO projects NIDFORVAL (ID #28607, PI G. Pinardi, BIRA-IASB) and CESAR (ID #28596, PI A. Apituley, KNMI). S. Compernolle, G. Pinardi and T. Verhoelst at BIRA-IASB acknowledge national funding from BELSPO and ESA through the ProDEx projects TROVA-E2 (PEA 4000116692). The authors express special thanks to A.M. Fjæraa, J. Granville, S. Niemeijer and O. Rasson for post-processing of the network and satellite data and for their dedication to the S5p operational validation.

This work contains modified Copernicus Sentinel-5 Precursor satellite data (2018-2020) processed by KNMI and post-processed by BIRA-IASB. The ZSL-DOAS and Pandora data used in this publication were obtained as part of the Network for the Detection of Atmospheric Composition Change (NDACC, https://ndacc.org) and the Pandonia Global Network (PGN, https://www.pandonia-global-network.org/), respectively, and are publicly available. The LATMOS Real-Time processing facility is acknowledged for fast delivery of ZSL-DOAS SAOZ data. Fast delivery of MAX-DOAS data tailored to the S5p validation was organized through the S5PVT AO project NIDFORVAL. The authors are grateful to ESA/ESRIN for supporting the ESA Validation Data Centre (EVDC) established at NILU, and for running the Fiducial Reference Measurements (FRM) programme and in particular the FRM4DOAS and Pandonia projects.

The MAX-DOAS, ZSL-DOAS and PGN instrument PIs and staff at the stations are thanked warmly for their sustained effort on maintaining high quality measurements and for valuable scientific discussions. A. Elokhov and A. Gruzdev acknowledge national funding from RFBR through the project 20-05-274. IUP-Bremen acknowledges DLR-Bonn for funding received through project 50EE1709A. The SAOZ network acknowledges funding from the French Institut National des Sciences de l'Univers (INSU) of the Centre National de la Recherche Scientifique (CNRS), Centre National d'Etudes Spatiales (CNES) and Institut polaire français Paul Emile Victor (IPEV). Work done by HI was supported by the Environment Research and Technology Development Fund (2-1901) of the Environmental Restoration and Conservation Agency of Japan, JSPS KAKENHI (grant numbers JP19H04235 and JP17K00529), the JAXA 2nd research announcement on the Earth Observations (grant number 19RT000351), and JST CREST (grant number JPMJCR15K4). The U Toronto ZSL-DOAS measurements at Eureka were made at the Polar Environment Atmospheric Research Laboratory (PEARL) by the Canadian Network for the Detection of Atmospheric Change (CANDAC), with support from the Canadian Space Agency (AVATARS project), the Natural Sciences and Engineering Research Council (PAHA project), and Environment and Climate Change Canada.





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
