# Peer review of "Ground-based validation of the Copernicus Sentinel-5p TROPOMI NO2 measurements with the NDACC ZSL-DOAS, MAX-DOAS and Pandonia global networks"

_Atmospheric Measurement Techniques, 2020_

## Referee Comment (RC1) · Anonymous Referee #1 · 10 Jun 2020

This paper thoroughly covers the various aspects of comparisons with both MAX-DOAS and Direct-Sun ground-based instruments. In addition, it discusses differences with a previous data set derived from OMI. All the algorithms are described or referenced. The paper is well written and easy to understand if some time is spent carefully reading through all the abbreviations. The figures and their meaning are clear. The paper will serve as a good reference paper for TROPOMI details in future science papers.

A consistent theme in the validation process is the underestimate of column NO2 com-

pared to ground-based measurements, both MAX-DOAS and direct sun. The main effect causing the differences is area averaging over the TROPOMI pixel compared to the very local observations from ground-based instruments. Agreement when pollution effects are small or zero is quite good because the stratospheric component of NO2 is much more spatially homogeneous. The disagreement increases as the pollution level increases along with spatial inhomogeneity. In the present document, the authors treat the spatial averaging effect as uncertain. A comparison of TROPOMI with the larger OMI area averaging effect from its larger pixel size should be convincing. The paper should include a stronger statement about the effect of area averaging on ground-based validation of TROPOMI. Line 27: nitrates, which are Line 30: local national regulations limiting boundary Line 47: on a global scale Line 51: Onwards Line 109 processor versions to which this corresponds Line 294 the referenced site does not contain all the data that were used in this paper. This should be fixed

---

## Referee Comment (RC2) · Anonymous Referee #2 · 7 Jul 2020

The manuscript titled "Ground-based validation of the Copernicus Sentinel-5p TROPOMI NO2 measurements with the NDACC ZSL-DOAS, MAX-DOAS and Pandonia global networks" provides a comprehensive validation information of Total, Tropospheric, and stratospheric TROPOMI NO2 data based on three types of the ground based remote sensing data. The manuscript contains the scientific information, which is thought to contribute to the community of this field for sure. The paper is well written and easy to follow. However, there are a few things that need to be addressed and discussed to enhance the quality of the paper.

Specific comments: Lines 150-155: (1) Please address the accuracy of the stratospheric NO2 column retrieved from the ZSL-DOAS. (2) Temporal resolution of the ZSL-DOAS data and differences of measurement time between the ZSL-DOAS and TROPOMI needs to be discussed. (3) Spatial coverage of the ZSL-DOAS data need also to be specified.

Line 184: Please specify how large footprint which tropospheric NO2 are averaged over. It can be a specific area size or a range of the area sizes. It will help the readers quantitatively understand the horizontal representativeness of the stratospheric NO2 column from the ZSL-DOAS.

Line 189: "A small negative bias": I recommend not to use "bias" unless ZSL DOAS accuracy is proven to be much higher than that of TROPOMI or space borne UV hyperspectral sensors.

Lines 211-220: The manuscript addresses that there is an issue of 10% overestimation of the PGN NO2 data at high altitude stations due to using cross sections at a single temperature. Please consider removing the Section 3.4 since of the PGN NO2 data at high altitude stations is not accurate enough for validating stratospheric NO2 from TROPOMI as the authors also mention it.

Figure 7: Y Axis: (1) Why using SAT-GND? All other figures use "TROPOMI". For consistency, please consider using something like "TROPOMI-ZSLDOAS" or anything better. (2) Please enlarge the figure and explain what the colors represent in the caption?

Section 4.1 and 4.3: (1) Authors need to address quantitative differences of tropospheric NO2 columns retrieved from various MAX-DOAS instruments and their algorithms. (2) Errors and accuracy of the retrieved tropospheric NO2 column needs to be both quantitatively and qualitatively addressed before discussing comparison results in Section 4.3.

Lines: 264-280: (1) Please address major factors that cause the difference between tropospheric NO2 column data obtained from MAX-DOAS and TROPOMI. (2) Please discuss the possible reason for larger discrepancy at more polluted sites. I personally think one of the things that authors need to do is to compare aerosol properties and aerosol extinction profiles used to retrieve tropospheric NO2 column between MAX-DOAS and TROPOMI.

Section 5 Total column validation: Is there any problem associated with cross section at a single temp.? Please clarify it since there is the issue at Section 3.

———————————————————

---

## Author Comment (AC1) · 19 Aug 2020

Dear referee,

Many thanks for your careful reading of our manuscript, and for the valuable feedback. We address your concerns below (and in a modified version of the manuscript).

Referee comment: A consistent theme in the validation process is the underestimate of column NO2 compared to ground-based measurements, both MAX-DOAS and direct sun. The main effect causing the differences is area averaging over the TROPOMI pixel compared to the very local observations from ground-based instruments. Agreement when pollution effects are small or zero is quite good because the stratospheric component of NO2 is much more spatially homogeneous. The disagreement increases as the pollution level increases along with spatial inhomogeneity. In the present document, the authors treat the spatial averaging effect as uncertain. A comparison of TROPOMI with the larger OMI area averaging effect from its larger pixel size should be convincing. The paper should include a stronger statement about the effect of area averaging on ground-based validation of TROPOMI.

**Answer:** Indeed, the underestimation of tropospheric columns (and of total columns when these contain a significant tropospheric contribution) in the S5p product is a clear outcome of our comparisons, which is fully in line with the outcomes of other comparable validation exercises, both on S5p and on other satellite NO2 data sets. While differences in area averaging most certainly contribute to this, and while this may have been the dominant effect in the underestimation by larger-footprint sounders such as GOME, GOME-2, and OMI, we are less certain it is the dominant cause in our S5p comparisons, for the following reasons:

- The S5p pixel size is much smaller than that of the other sounders, and it is comparable to larger emission sources (cities, harbours,...). The footprint is still (much) larger than that of the MAX-DOAS or Pandora, but unless the latter is really positioned near the peak of the emission source, the difference in area averaging can work both ways (i.e. not necessarily leading to an underestimation, but mostly increasing the scatter).
- In a preceding study on OMI vs. MAX-DOAS and Pandora (Compernolle et al. 2020), we already concluded that area-averaging can not be the sole cause, and that short-comings in the assumed vertical profile, in particular in polluted conditions, probably is a key effect. This is corroborated, for S5p, by several studies demonstrating the reduction in negative bias by replacing the a priori profile with one taken from a more detailed regional

model (e.g. Ialongo et al., AMT., 13, 205–218, 2020; Tack et al., AMTD 2020-148). While this is to some extent also a horizontal resolution effect (of the underlying profile climatology/model), it is not an NO2 area averaging effect in itself.

- A similar effect is found for assumed aerosol concentrations (Liu et al, AMTD 2019-500)
- An upcoming improved cloud product, a key input to the NO2 retrieval, has already been demonstrated to reduce the underestimation (Eskes et al, in prep.).

A comparison between OMI and TROPOMI comparisons should indeed show the impact (i.e. a stronger underestimation) of the larger area averaging of OMI, but it can be argued (as done above) that this does not imply a similar scale issue for TROPOMI vs. MAX-DOAS or Pandora (because TROPOMI is starting to resolve the emission sources, which is much less the case for OMI). We have now mentioned the effect of area averaging explicitly in the abstract, and it is put in perspective (along the same lines as described above) starting at line 389.

Line 27: nitrates, which are -> fixed Line 30: local national regulations limiting boundary -> fixed Line 47: on a global scale -> fixed Line 51: Onwards -> fixed Line 109: processor versions to which this corresponds -> fixed Line 294: the referenced site does not contain all the data that were used in this paper. -> These 2 websites (EVDC and PGN) should together contain all the data, as that is where we obtained them. Please provide us with more specifics if something is missing on these archives.

---

## Author Comment (AC2) · 19 Aug 2020

Dear referee,

Many thanks for your valuable feedback on our paper. We address your concerns below (and in a modified version of the manuscript).

Lines 150-155: 1) Please address the accuracy of the stratospheric NO2 column retrieved from the ZSL-DOAS. (2) Temporal resolution of the ZSL-DOAS data and Printer-friendly version

differences of measurement time between the ZSL-DOAS and TROPOMI needs to be discussed. (3) Spatial coverage of the ZSL-DOAS data need also to be specified. **Answer:** This information is presented in the manuscript in lines 164 to 173 (geographical distribution and accuracy), and in the Section thereafter (differences in measurement time and horizontal sensitivity). For the exact dimensions of the footprint, see our answer to the following comment.

**Line 184:** Please specify how large footprint which tropospheric (we assume stratospheric is meant) NO2 are averaged over. It can be a specific area size or a range of the area sizes. It will help the readers quantitatively understand the horizontal representativeness of the stratospheric NO2 column from the ZSL-DOAS.

**Answer:** We added at line 187 that the length of this footprint if of the order of 300-600 km in the direction of the sun, and the width is typically of the order of 50-100 km at mid latitudes, depending on the duration of sunrise and sunset.

**Line 189:** "A small negative bias": I recommend not to use "bias" unless ZSL DOAS accuracy is proven to be much higher than that of TROPOMI or space borne UV hyperspectral sensors.

**Answer:** Agreed, we replaced "bias" with "median difference", also in the part of the discussion related to the stratospheric columns.

**Lines 211-220:** The manuscript addresses that there is an issue of 10% overestimation of the PGN NO2 data at high altitude stations due to using cross sections at a single temperature. Please consider removing the Section 3.4 since of the PGN NO2 data at high altitude stations is not accurate enough for validating stratospheric NO2 from TROPOMI as the authors also mention it.

**Answer:** The use of cross sections at a single (tropospheric) temperature in the PGN data processing indeed deserves a clear caveat. However, as the effect is "only" of the order of 10%, we believe it still makes sense to show the results: With or without a hypothetical 10% correc-
tion, they independently confirm that TROPOMI stratospheric columns are not severely biased. We have added another explicit caveat on this issue in the PGN data presentation in Section 5.1.

**Figure 7:** Y Axis: (1) Why using SAT-GND? All other figures use "TROPOMI". For consistency, please consider using something like "TROPOMI-ZSLDOAS" or anything better. (2) Please enlarge the figure and explain what the colors represent in the caption?

Answer: This is fixed in a new version of the figure.

**Section 4.1 and 4.3:** (1) Authors need to address quantitative differences of tropospheric NO2 columns retrieved from various MAX-DOAS instruments and their algorithms. (2) Errors and accuracy of the retrieved tropospheric NO2 column needs to be both quantitatively and qualitatively addressed before discussing comparison results in Section 4.3.

**Answer:** Information on the retrieval methods used for the different MAX-DOAS data sets has been added in table A2 and, while the assessment of the differences in tropospheric NO2 VCD due to the use of MAX-DOAS spectrometers with various instrumental performance levels and different retrieval algorithms, and hence different systematic and random uncertainty sources, is complex, the following discussion is added to Sect. 4.1 (including a new figure added to the supplement and copied at the bottom of this Author Comment):

Published total uncertainty estimate on the NO2 tropospheric VCD are of the order of 7-17% in polluted conditions, including both random (around 3 to 10% depending on the instrument) and systematic (11 to 14%) contributions (Irie et al., 2008; Wagner et al., 2011; Hendrick et al., 2014; Kanaya et al., 2014). These ranges are more or less confirmed by the uncertainties reported in the data files, as visualized in Fig A.1 in the supplement. Nevertheless, differences in the reported uncertainties and in the actual measurement of the same scene between individual instruments are sometimes larger and the main potential sources of these inhomogeneities are listed below:

AMTD
- Different uncertainty reporting strategy: the reported systematic uncertainty may include only that from the NO2 cross sections (approx. 3%; UNAM, BIRA-IASB, MPIC, AUTH, IUPB) or it may include also a contribution from the VCD retrieval step (up to 14% in JAMSTEC data and 20% in KNMI data).
- Different SCD retrieval: Recommended common DOAS settings are used by all groups in the present study, and if doing so, instrument intercomparison campaigns like CINDI-1 and -2 (Roscoe et al., 2010; Kreher et al., 2020) revealed relative biases between 3 and 10% in DSCD.
- Different VCD retrieval methods: Using either (1) vertical profile inversion using optimal estimation (BIRA-IASB, UNAM), (2) profile inversion using parameterized profile shapes (JAMSTEC and ChibaU), (3) direct retrieval via the calculation of a tropospheric AMF (QA4ECV datasets), or (4) direct retrieval using a geometrical approximation, can lead to systematic differences in the 5-15% range (Vlemmix et al., 2015b, and Friess et al., 2019).

Consequently, expert judgment on the total uncertainty at the network level yields a conservative estimate of 30% uncertainty in polluted conditions. Ongoing efforts to harmonise MAX-DOAS tropospheric NO2 data processing, e.g. as part of the ESA FRM4DOAS project, should help minimizing such network inhomogeneities in the near future.

**Lines: 264-280:** (1) Please address major factors that cause the difference between tropospheric NO2 column data obtained from MAX-DOAS and TROPOMI. (2) Please discuss the possible reason for larger discrepancy at more polluted sites. I personally think one of the things that authors need to do is to compare aerosol properties and aerosol extinction profiles used to retrieve tropospheric NO2 column between MAXDOAS and TROPOMI.

Answer: An extensive discussion of known and potential causes for the discrepancy at polluted
sites is indeed only provided (much) further on in the manuscript (in Section 6, near line 360). We have entered a reference to this discussion section at line 281.

Concerning the impact of the aerosols properties: This is difficult to assess qualitatively as only a subset of the MAX-DOAS stations report the AOD used in the retrieval (and this can be the one coming from the O4 analysis, from an AOD climatology as for the QA4ECV cases, or from co-located AERONET instruments), and in the TROPOMI files, only the aerosol\_index\_354\_388 information is provided. Still, we agree it requires further discussion in the manuscript. Consequently, this possible source of discrepancies is now discussed in more detail in the 2nd bullet point in the Discussion. In particular, the following information was added:

Somewhat related to the vertical sensitivity is the treatment of aerosol optical depth and its vertical profile. Poor representation of the aerosol opacity has been shown (from simulations) to cause both underestimated NO2 in satellite retrievals and overestimated NO2 in MAX-DOAS measurements (Leitao et al., 2010; Ma et al., 2013; Jin et al., 2016). Satellite-ground discrepancies in previous validation studies have already been attributed to such aerosol issues (Boersma et al., 2018; Compernolle et al., 2020). Moreover, explicit aerosol corrections in the S5p retrievals have already been shown to improve the agreement (Liu et al., 2020).

**Section 5:** Total column validation: Is there any problem associated with cross section at a single temp.? Please clarify it since there is the issue at Section 3.

**Answer:** Very pertinent point. The results at "clean sites" should indeed be interpreted with care as the PGN data are believed to be overestimated here by approx. 10%. This would suggest an actual positive mean difference (bias) for TROPOMI of similar size when little pollution is present, i.e. when the column is mostly stratospheric. Such a statement was added to the paper near line 340 (besides the clear caveat already formulated at the introduction of the PGN data). Note that this is somewhat at odds with the slight negative mean difference found in the ZSL-DOAS comparisons and probably reflects the true accuracy of the ground-based data, which should thus be taken to be of the order of +/- 10% at best.
Fig. 1.

---

## Author Response (AR2)

Dear referee,

Many thanks for your valuable feedback on our paper. We address your concerns below (and in a modified version of the manuscript).

**Cross sections:** Please compare the NO2 absorption cross sections used for the NO2 retrievals from MAX-DOAS, ZSL-DOAS, Pandora, and S5 using a table and summarize the differences in magnitude and its quantitative effects on the NO2 retrieval.

*Answer: Such a table has been added to the paper at the end of Sect. 2. (i.e. the section describing the data sets), including some discussion. In short: Most products use the cross sections published by Vandaele et al. (1998), but there are differences in the choice of temperature at which to take the cross sections. The ZSL-DOAS measurements are processed with cross sections at a fixed 220K or 227K, i.e. typical stratospheric temperatures. MAX-DOAS data are processed either with cross sections at room temperature (298K, representing a typical tropospheric temperature) or using an orthogonalized set of cross sections at 298K and 220K when both tropospheric and stratospheric slant columns are retrieved. As the scientific focus of the PGN up until processor version 1.7 (used for this study) was on measuring polluted conditions, i.e. in the presence of moderate to large tropospheric columns, the cross sections used in the processor are scaled to a fixed effective temperature of 254.4K, which corresponds to the situation of approximately equal column amounts in the troposphere and stratosphere. The S5p retrievals use cross sections at 220K, but with an explicit correction for the temperature dependence of the NO2 cross sections in the AMF: Space-time co-located daily ECMWF temperature profile forecasts are used to compute a height-dependent AMF correction factor. The temperature sensitivity parametrized in this correction is approximately 0.32%/K (Zara, 2017). A posteriori temperature correcting the ground-based data is beyond the scope of this paper, so it must be kept in mind that this may contribute to differences between S5p and ground-based columns. Specifically, we could expect a small seasonal cycle in the stratospheric column comparisons of a few percent due to the seasonal variation in stratospheric temperature not being accounted for in the ZSL-DOAS data processing. PGN columns may either be overestimated by up to 10% when the column is mostly stratospheric or underestimated by a similar order of magnitude when large tropospheric amounts are present. The MAX-DOAS data may be biased in either direction by a few percent when tropospheric and/or stratospheric temperatures differ strongly from the 298K and 220K default temperatures.*

**The errors of the ground data:** For the validation of the S5p, the uncertainties and errors of the NO2 products retrieved from the ground based MAX-DOAS, ZSL-DOAS, and Pandora should be addressed in detail and summarized in a table.

**Answer:** *We have complemented the textual discussion of uncertainties in the different ground-based data products with a summarizing table (Table 2) near the beginning of Sect. 2. To link this table and the discussion on the cross sections more directly to the description of the ground-based data sets, we have restructured the paper somewhat: The data sets are now presented together in Sect. 2, and no longer as part of the individual validation sections.*

**In Figure 7:** please make the dot size smaller, so that readers can distinguish their locations better.
**Answer:** *A new version of this graph, with smaller markers, was produced and included.*

**Lines 175- 180:** The ZSL-DOAS measurements at sunset are adjusted to the early-afternoon S5p overpass time using a model-based correction factor. Thus, it is important for readers to understand the uncertainties of PPSCBOX 1D stacked-box photochemical model (Errera and Fonteyn, 2001; Hendrick et al., 2004). Please address the uncertainty of its simulated diurnal cycle.
**Answer:** *We have added a discussion on the uncertainty of this correction in the manuscript (near line 180). It reads as follows: This photochemical correction factor is an average based on ten years of the box-model simulations, and the range of values over these 10 years can be considered an uncertainty estimate. It varies between 1% and 6% at the sites considered here, the uncertainty being largest at high latitudes in local winter. This does however not contain any model uncertainty (in the sense of the accuracy of the model in representing the true photochemical variation during the day). Another way to estimate the uncertainty in the adjusted ZSL-DOAS data is by comparing the agreement between sunrise and sunset measurements when both are photochemically adjusted to the S5p overpass time. This does also contain co-location mismatch uncertainty due to transport of air occuring during the period between sunrise and sunset, and due to the different airmasses that are probed (East or West of the instrument respectively). Moreover, it also contains that part of the measurement uncertainty that is not systematic on a daily (or longer) timescale. We find that sunrise and sunset measurements typically agree within 6% (standard deviation of the differences).*

**Validation methods:** The treatment of aerosol optical depth and its vertical profile is important for the S5p NO2 quality. To sophisticated validations for aerosol and cloud effects on discrepancy between the TROPOMI NO2 and those ground based data, consider additional comparisons of the S5p with those three ground based measurements in terms of AOD levels and cloud fraction. Check if the comparisons can be made in terms of aerosol peak eight as well.
**Answer:** *We agree with the referee on the importance of a correct AOD and cloud treatment for good data retrievals, both ground-based and satellite. We*

have added graphs that present the dependence of the differences in tropospheric columns on (1) the AOD retrieved in the MAX-DOAS retrieval, and (2) the cloud radiance fraction as used in the S5p NO2 retrieval. The dependence of the stratospheric column differences on cloud fraction is presented in the middle panel of Fig. 7. No clear dependence of the bias on either property is seen, though in view of the relatively large scatter in the tropospheric column comparison, this does not preclude more subtle dependencies. The impact of aerosol peak height was impossible to judge within the scope of the current paper as no such information is readily available.

[revised manuscript text omitted]

**2.3 MAX-DOAS data**

Satellite tropospheric $NO_2$ column data are compared clasically to correlative measurements acquired by MultiAxis-Differential Optical Absorption Spectroscopy (MAX-DOAS) instruments (Hönninger and Platt, 2002; Honninger et al., 2004; Sinreich et al., 2005). MAX-DOAS instruments measure from sunrise to sunset the UV-visible radiance scattered in several directions and elevation angles, from which the tropospheric VCD and/or the lowest part of the tropospheric $NO_2$ profile (usually up to 3km altitude, and up to 10km at best) can be retrieved through different techniques

**Table 2.** Estimated uncertainties for the different types of ground-based measurements used in this work. Ex-ante refers to uncertainties provided with the data, based on a propagation of raw measurement uncertainties and on sensitivity analyses. Ex-post refers to uncertainty estimates derived by comparison with other (independent) measurements, which inevitably also contain some representativeness uncertainties. More detail is provided in the dedicated subsections of Sect. 2.

[revised manuscript text omitted]

215   $NO_2$ data (see Sect. 4).

**2.5  $NO_2$ cross section data**

A potential source of inconsistencies between the different data products lies in the $NO_2$ cross sections that are used. An overview of the different choices made is provided in Table 3. Most products use the cross sections published by Vandaele et al. (1998), but there are differences in the choice of temperature at which to take the cross sections. The ZSL-DOAS

220  measurements are processed with cross sections at a fixed 220K or 227K, i.e. typical stratospheric temperatures. MAX-DOAS data are processed either with cross sections at room temperature (298K, representing a typical tropospheric temperature) or using an orthogonalized set of cross sections at 298K and 220K when both tropospheric and stratospheric slant columns are retrieved. As the scientific focus of the PGN up until processor version 1.7 (used for this study) was on measuring polluted conditions, i.e. in the presence of moderate to large tropospheric columns, the cross sections used in the processor are scaled

225  to a fixed effective temperature of 254.4K, which corresponds to the situation of approximately equal column amounts in the troposphere and stratosphere. The S5p retrievals use cross sections at 220K, but with an explicit correction for the temperature dependence of the $NO_2$ cross sections in the AMF: Space-time co-located daily ECMWF temperature profile forecasts are used to compute a height-dependent AMF correction factor. The temperature sensitivity parametrized in this correction is approximately 0.32%/K (Zara et al., 2017). A posteriori temperature correcting the ground-based data is beyond the scope

230  of this paper, so it must be kept in mind that this may contribute to differences between S5p and ground-based columns. Specifically, we could expect a small seasonal cycle in the stratospheric column comparisons of a few percent due to the seasonal variation in stratospheric temperature not being accounted for in the ZSL-DOAS data processing. PGN columns may either be overestimated by up to 10% when the column is mostly stratospheric or underestimated by a similar order of magnitude when large tropospheric amounts are present. The MAX-DOAS data may be biased in either direction by a few

235  percent when tropospheric and/or stratospheric temperatures differ strongly from the 298K and 220K default temperatures.

**Table 3.** $NO_2$ cross section source and temperature for the different data processings used in this work. More detail is provided in Sect. 2.5.

| Instrument | reference | temperature | comments |
|---|---|---|---|
| S5p TROPOMI | Vandaele et al. (1998) | 220K | With temperature correction in AMF (Zara et al., 2017) |
| ZSL-DOAS | Vandaele et al. (1998) | 220K | |
| ZSL-DOAS | Harder et al. (1997) | 227K | NIWA instruments |
| MAX-DOAS | Vandaele et al. (1996) | 298K | tropospheric retrieval only |
| MAX-DOAS | Vandaele et al. (1998) | 298K and 220K | Orthogonalized following Peters et al. (2017) |
| PGN | Vandaele et al. (1998) | 254.4K | PGN processor v1.7 |

**3  Mutual coherence between TROPOMI NRTI and OFFL**

[revised manuscript text omitted]

310 over these 10 %, years can be considered an uncertainty estimate. It varies between 1% and 6% at the sites considered here, the uncertainty being largest at high latitudes in local winter. This does however not contain any model uncertainty (in the sense of the accuracy of the model in representing the true photochemical variation during the day). Another way to estimate the uncertainty in the adjusted ZSL-DOAS data is by comparing the agreement between sunrise and sunset measurements

when both are photochemically adjusted to the S5p overpass time. This does also contain co-location mismatch uncertainty due to transport of air occuring during the period between sunrise and sunset, and due to the different airmasses that are probed (East or West of the instrument respectively). Moreover, it also contains that part of the measurement uncertainty that is not systematic on a daily (or longer) timescale. We find that sunrise and sunset measurements typically agree within 6% (standard deviation of the differences). Overall, the ~~main source of uncertainty probably being the effective SZA to assign to the full twilight measurement period. To reduce mismatch errors due to the significant difference in horizontal sensitivity between S5p and ZSL-DOAS measurements , individual TROPOMI stratospheric column data (in ground pixels at high horizontal sampling) are averaged over the much larger footprint of the air mass to which the ground-based zenith-sky measurement is sensitive, see Lambert et al. (1997b, ISBN 978-1-4614-3908-0, © Springer New York, 2012) and Verhoelst et al. (2015) for details. Note that , as the TROPOMI stratospheric column is a TM5 output, it's true resolution is actually much lower than the pixel size.~~ 10%-14% total uncertainty estimate already presented in Sect. 2.2 thus seems realistic.

**4.2 Comparison results**

[revised manuscript text omitted]

**6**

**5.1**

 Two key influence quantities for observations of tropospheric NO$_2$

[Figure]

**Figure 10.** Same as Fig. 6, but now for the difference between S5p TROPOMI OFFL and MAX-DOAS NO₂ tropospheric columns, and with ordered as a function of the median ground-based tropospheric column (largest median VCD values on top). The line represents the median difference. Box bounds represent 25 and 75 percentile, while whiskers indicate the 9 and 91 percentiles. The shaded area corresponds to the mission requirement of maximum 50% for the bias.

430

For the current work, 25 sites have contributed Pandora data, collected either from the ESA Atmospheric Validation Data Centre (EVDC) () or from the PGN data archive (). Only data files from a recent quality upgrade (processor version 1.7, retrieval version nvs1, with file version 004 and 005; see ) were used, with 005 files (consolidated data) having precedence over 004 files (rapid delivery data are aerosol optical depth (AOD) and cloud (radiance) fraction (CRF). The most important change with the previous data release is a more stringent quality filtering. Seventeen sites have provided measurement data newer than 3 months.

Except at low sun elevation, the footprint of these direct-sun measurements is much smaller than a TROPOMI pixel. Therefore, - as it is the case with dependence of the differences between MAX-DOAS - a significant horizontal smoothing difference error can be expected in the TROPOMI-Pandora comparison, especially in and TROPOMI tropospheric columns on these two influence quantities is visualized in Fig. 11. AOD is only retrieved in the processing of a handful of MAX-DOAS instruments, the case of tropospheric gradients and when tropospheric is the largest contributor to the total column . others using climatological information, hence the limited subset in stations in the upper panel of this figure. No clear dependence of the bias on either property is seen, though in view of the relatively large scatter in these tropospheric column comparisons, this does not preclude more subtle dependencies. The impact of aerosol peak height would also be interesting to assess, but this is impossible to judge within the scope of the current paper as no such information is readily available.

Three Pandora instruments (Altzomoni, Izaña, Mauna Loa) are located near the summit of a volcanic peak and are therefore not sensitive to the lower-lying tropospheric . In this work, their observations are compared to the TROPOMI stratospheric data (see Sect. 4).

**6 Total column validation**

**6.1 Filtering, co-location and harmonization**

As was done for the tropospheric column validation in Sect. 5, only S5p pixels with $\mathrm{qa\_value}$ at least 0.75 are retained. The so-called summed product is used, i.e. the total column computed as the stratospheric plus the tropospheric column values. This summed column differs from the total column product. Only Pandonia measurements with the highest quality label (0 and 10) are used. The average column value within a 1-hour time interval, centered on the S5p overpass time, is used. As the $NO/NO_2$ ratio varies only slowly around the afternoon solar local time of the TROPOMI overpass, this small temporal window ensures no model-based adjustment is required. A 30-minute time interval was tested as well, but this did not change significantly the results. Moreover, only TROPOMI pixels containing the station were considered.

**6.2 Comparison results**

An example of a time series of co-located TROPOMI and PGN total column measurements, and their difference, is shown in Fig. 12.

[Figure]

**Figure 11.** Dependence of the difference between TROPOMI OFFL and ground-based MAX-DOAS tropospheric $NO_2$ column data on the MAX-DOAS retrieved aerosol optical depth (AOD, upper panel, only available for a subset of the instruments) and satellite cloud radiance fraction (CRF, bottom panel).

[revised manuscript text omitted]

850    M., Piters, A., Remmers, J., Wang, Y., Wagner, T., Wang, S., Saiz-Lopez, A., García-Nieto, D., Cuevas, C. A., Benavent, N., Querel, R., Johnston, P., Postylyakov, O., Borovski, A., Elokhov, A., Bruchkouski, I., Liu, H., Liu, C., Hong, Q., Rivera, C., Grutter, M., Stremme, W., Khokhar, M. F., Khayyam, J., and Burrows, J. P.: Investigating differences in DOAS retrieval codes using MAD-CAT campaign 
[revised manuscript text omitted]
., Hermans, C., Simon, P., Carleer, M., Colin, R., Fally, S., M'erienne, M., Jenouvrier, A., and Coquart, B.: Measurements of the NO2 absorption cross-section from 42 000 cm-1 to 10 000 cm-1 (238-1000 nm) at 220 K and 294K, J. Quant. Spectrosc. Radiat. Transfer, 59, 171–184, https://doi.org/10.1016/s0022-4073(97)00168-4, 1998.

Vandaele, A. C., Hermans, C., Simon, P. C., Van Roozendael, M., Guilmot, J. M., Carleer, M., and Colin, R.: Fourier transform measurement of NO2 absorption cross-section in the visible range at room temperature, Journal of Atmospheric Chemistry, 25, 289–305, https://doi.org/10.1007/BF00053797, https://doi.org/10.1007/BF00053797, 1996.

[revised manuscript text omitted]